# Projected changes in atmospheric moisture transport contributions associated with climate warming in the North Atlantic

José C. Fernández-Alvarez [1,2], Albenis Pérez-Alarcón [1,2], Jorge Eiras-Barca[1,3], Stefan Rahimi[4,5], Raquel Nieto [1] & Luis Gimeno [1] ✉

Global warming and associated changes in atmospheric circulation patterns are expected to alter the hydrological cycle, including the intensity and position of moisture sources. This study presents predicted changes for the middle and end of the 21st century under the SSP5-8.5 scenario for two important extratropical moisture sources: the North Atlantic Ocean (NATL) and Mediterranean Sea (MED). Changes over the Iberian Peninsula—considered as a strategic moisture sink for its location—are also studied in detail. By the end of the century, moisture from the NATL will increase precipitation over eastern North America in winter and autumn and on the British Isles in winter. Moisture from the MED will increase precipitation over the southern and western portions of the Mediterranean continental area. Precipitation associated with the MED moisture source will decrease mainly over eastern Europe, while that associated with the NATL will decrease over western Europe and Africa. Precipitation recycling on the Iberian Peninsula will increase in all seasons except summer for mid-century. Climate change, as simulated by CESM2 thus modifies atmospheric moisture transport, affecting regional hydrological cycles.

It is predicted that thermal warming of the atmosphere will exceed the 1.5 °C or 2 °C targets in the 21st century unless steep reductions in $CO_2$ and other greenhouse gas emissions are made in the coming decades[1]. The thermodynamic response to a warmed atmosphere per degree of warming is a 6–7% increase in low-level atmospheric water vapour[2], according to the Clausius–Clapeyron relationship[3–5], and this will have a strengthening impact on moisture transport worldwide[5]. In addition to warming, an increase in global mean annual evaporation is expected[5], affecting the evaporation rates of most oceans[6]. Moisture transport will also increase[2,7,8] and so will the vertically integrated water vapour transport (IVT)[9]. It has been shown that this effect dominates over circulation changes in the mid-latitudes[10]. Moreover,

changes in the atmospheric circulation patterns will also occur as the planet retains more heat, which will affect the atmospheric water balance at global and regional levels[11].

The net effect of dynamic (circulation) and thermodynamic processes is therefore extremely relevant when analysing future changes in moisture sources for precipitation over a region[12]. It is estimated that continental moisture recycling will decrease by 2–3% per °C globally[13], being systematically higher in the past and lower in the future[14]. However, there will be exceptions: an increase of ~2–8% is expected in West Africa and the Iberian Peninsula[14] by the end of the 21st century. Thus, the projected decrease in global recycling, coupled with the inherent moisture limitations of the land's surface, implies

[1]Centro de Investigación Mariña, Universidade de Vigo, Environmental Physics Laboratory (EPhysLab), Campus As Lagoas s/n, Ourense 32004, Spain. [2]Departamento de Meteorología, Instituto Superior de Tecnologías y Ciencias Aplicadas, Universidad de La Habana, La Habana, Cuba. [3]Defense University Center at the Spanish Naval Academy, Group of Applied Mathematics for Defense, Plaza de España s/n, 36920 Marín, Spain. [4]Department of Atmospheric Science, University of Wyoming, Laramie, WY 82071, USA. [5]Center for Climate Science, University of California Los Angeles, Los Angeles, CA 90095, USA. ✉e-mail: l.gimeno@uvigo.es

that the importance of oceans as land moisture sources will increase with warming[15].

It is thus essential to investigate the links between moisture sources and sinks, the role of climate change in modifying atmospheric moisture transport, and how this influences continental precipitation[16]. To our knowledge, previous studies based on Lagrangian approaches have not considered how climate change will alter the location and importance of moisture source regions and the future transport of moisture from such regions to continental areas.

Specifically, we analysed future changes in the moisture source–sink relationship around the Iberian Peninsula (IP), which is one of the hotspot mid-latitude areas affected by more than one moisture source[6]. This region is optimal for conducting a detailed study since its climate is highly dependent on the moisture intrusion from two of the major global oceanic sources[17,18] (Supplementary Fig. 1), the North Atlantic Ocean (NATL), main source of moisture for neighbouring continents in winter months[19,20] and the Mediterranean Sea (MED), known to be particularly sensitive to climate change and its implications[21,22]. The IP is also affected by strong recycling processes[20] (see Supplementary Section 1.3 for more information) and—along with the British Isles—it is the most active region in the North Atlantic for atmospheric river activity[23].

The scope of this study—although giving greater relevance to the Iberian Peninsula—reaches all the regions that have NATL and MED as a relevant source of moisture; North Africa, Western and Eastern Europe and Eastern North America. The analysis is performed using the Lagrangian dispersion model FLEXPART-WRF initialized with WRF-ARW (WRF with the dynamic core Advanced Research WRF−ARW−) dynamically downscaled outputs. Three representative climatic periods have been used for the comparison. Firstly, a historical period (HIST: 1985–2014), followed by a mid-century representative period (MC: 2036–2065), and finally, a late (end)-century representative period (EC: 2071–2100). The latter two have been selected under the SSP5-8.5 scenario of CMIP6[24,25] (see Data Description in Supplementary Section 1.1 and 'Methods'). The methodology used is a powerful tool for studying regional climate processes, by providing high-resolution historical and future climate data. It will allow us to gain better insights of future changes in the moisture source-sink relationship in the study area[26].

The results presented herein, relying on projected climate conditions under a warming atmosphere throughout the 21st century, reveal a general increase in oceanic moisture transport in the North Atlantic latitudes over its surrounding continental areas, an increase in the recycling processes for the Iberian Peninsula, and a projected decrease in the precipitation contribution from the MED mainly over Eastern Europe and from the NATL for Western Europe.

## Results

### Future projections in precipitation and geopotential height as per the CESM2 model

The study of changes in the moisture source regions must be contextualized within the framework of the general changes in the precipitation regime of the regions analysed. Thus, Supplementary Fig. 2 shows the seasonal and annual changes in the cumulative amount of precipitation (in mm/day) for both MC (2036–2065) and EC (2071–2100) with respect to the historical period (1985–2014) for the area of interest.

In annual terms, a decrease in precipitation close to 1 mm day$^{-1}$ is observed for EC over the entire Iberian Peninsula and the Mediterranean Sea. Decreases over 2 mm day$^{-1}$ are observed in the southwestern sector of the domain. On the other hand, increases close to 1 mm day$^{-1}$ are expected in almost the entire east coast of the USA and Canada, as well as in the Bermudas region. Specifically, the most notable increases in precipitation are observed in mid and high latitudes—as well as along the entire North American East Coast—for the winter months. Significant increases are also observed in the Gulf of Mexico region and

particularly in Bermudas for the autumn months. On the other hand, decreases in precipitation are characteristic of (sub)tropical latitudes for almost the entire year. In general terms, the patterns of precipitation changes are coincident between MC and EC; more accentuated in the latter.

Previous studies[27–29] already proposed a reduction in average precipitation for southern Europe. Specifically—and based on these previous studies—the expected precipitation reduction for the IP ranges from 10% to 15% for all seasons except winter. These changes would be in terms of a reduction in the number of wet days and an extension of drought periods by the end of the century[30].

It should be taken into account that a determining factor in the changes of the precipitation regime—and the associated sources of moisture—may be the variations in the atmospheric dynamics of the future climate, with regard to the present one. In order to infer these, a study analogous to the one shown for accumulated precipitation in Supplementary Fig. 2 is presented for the geopotential height at 500 hPa (Z500) in Supplementary Fig. 3.

In this regard, what is observed is a complex pattern, which can be summarized as a generalized increase of Z500 in the study region. The areas most affected by this increase are the subtropical regions, as well as the Atlantic coast of Canada and the northeastern USA. The areas least affected in annual terms—and particularly in the winter months—are the regions of usual influence of the Icelandic low. These results are in correspondence to those shown by Christidis et al.[31], where using seven climate models demonstrated that a significant global increase in the annual and seasonal mean Z500 is projected. Finally, it is noteworthy that this generalized—albeit moderate—increase in mid-level pressures for this particular region has been identified in the literature as a potential trigger for the easing of the westerly flow over the North Atlantic, which could affect the position of the jet stream located over this region[31].

### Future projections for integrated water vapour transport (IVT) in the North Atlantic Ocean

To better frame our assessment of the moisture transport processes, we also analysed changes in mean IVT fields within the extended area of the North Atlantic Ocean for MC and EC with respect to the historical pattern (Supplementary Fig. 4). The general trend (particularly for EC) is for an increase in IVT fields in the mid-latitudes of the North Atlantic, as well as in the Caribbean Sea and on the American east coast. In contrast, a decrease of roughly 40 kg m$^{-1}$ s$^{-1}$ is observed in the (sub) tropical latitudes of the North Atlantic. Intermediate seasons show transitional patterns: in spring, minimal changes are expected (with the exception of the East Caribbean Sea and Central Atlantic, where negative values are projected), and positive values are expected in autumn, mainly in the subtropical regions above 30°N.

Both MC and EC show similar patterns of change, being more accentuated in EC for almost all the regions. The only exception in this regard is observed on the European west coast in the summer months; where the increase is only observed for EC.

In addition, a northward shift of the maximum IVT fields is observed. This shift is coincident with the changes in precipitation already analysed in the previous section, and shown in Supplementary Fig. 2. These results are consistent with previous studies that have projected a poleward shift of subtropical high-pressure areas and of the frequent locations of atmospheric rivers[7]. A summary of the mean percentage differences with respect to the historical period is presented in Supplementary Table 2. In general, the maximum percentage values follow the Clausius–Clapeyron[3–5] relationship, showing an increase of ~7% K$^{-1}$.

### Changes in moisture sources for the Iberian Peninsula

The backward Lagrangian approach was used to evaluate future changes in the moisture sources for the IP, our featured target region

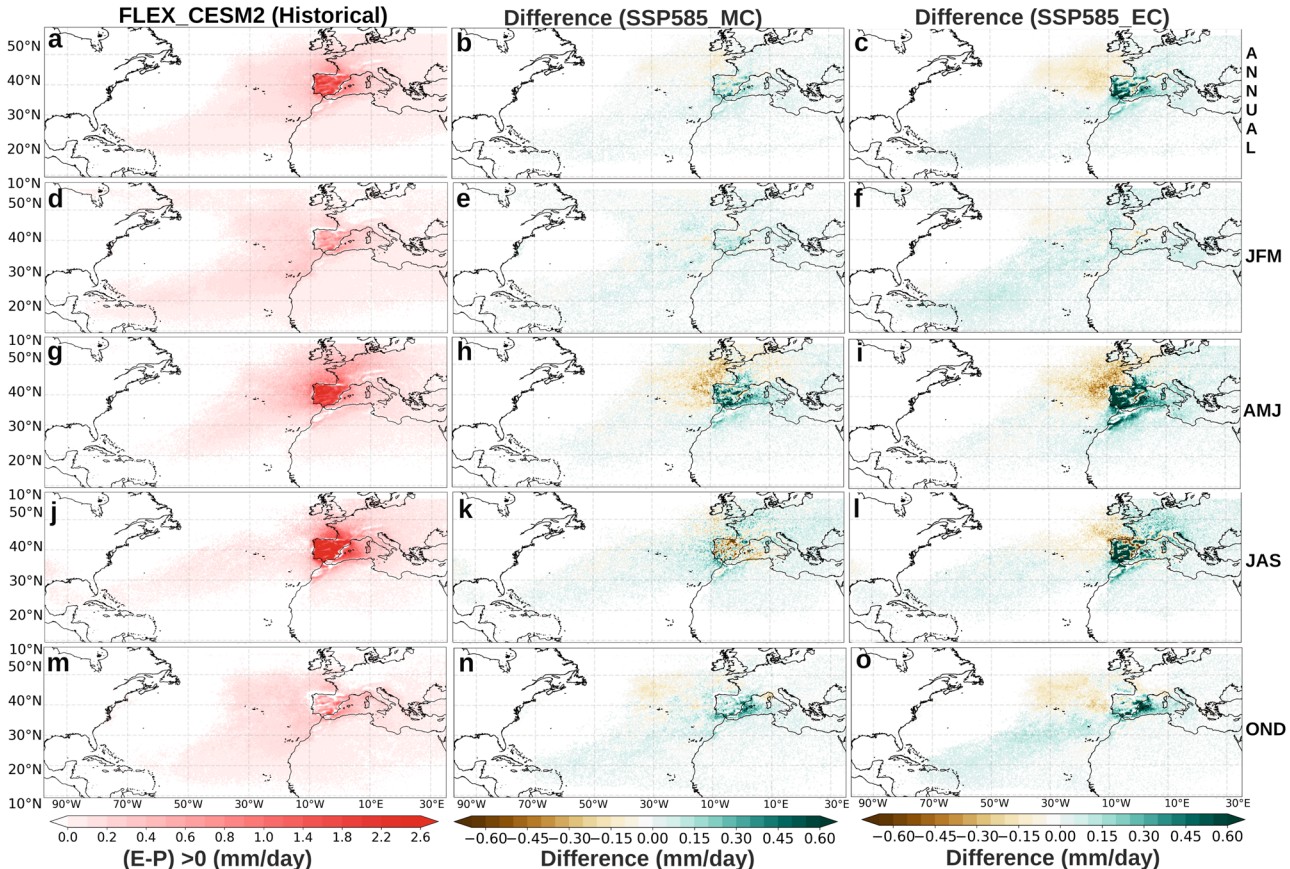

**Fig. 1 | Future changes in the absolute contribution (in mm day⁻¹) of the moisture sources to the Iberian Peninsula (IP).** Moisture sources fields (E-P > 0) for the IP in the historical reference period (1985–2014, (**a**, **d**, **g**, **j**, **m**)) and differences under the SSP5-8.5 scenario for the mid- and end 21st century (2036–2065, MC (**b**, **e**, **h**, **k**, **n**), and 2071–2100, EC (**c**, **f**, **i**, **l**, **o**), respectively) expressed in mm day⁻¹. The fields displayed from top to bottom correspond to annual, winter, spring, summer and autumn periods (ANNUAL, JFM, AMJ, JAS and OND).

(see 'Methods'). Figure 1 (left column) shows the moisture sources (E-P > 0) of the Iberian Peninsula during the historical period for CESM2.

In annual terms, it is observed that the precipitation recycling processes (PRPs)—processes that trigger the portion of precipitation whose origin lies in evaporation over the region[32]—have a practically homogeneous contribution in the whole Iberian Peninsula close to 1.4 mm/day.

In addition, this figure reveals known seasonal variations[20], with a greater contribution in winter from the North Atlantic reaching as far as the Caribbean Sea (-0.4–1.4 mm/day), and predominant influences from the Iberian Peninsula and the Mediterranean Sea in summer and spring (>1.8 mm/day). In autumn, the influence of oceanic sources and PRPs predominates in the east of the IP, with values that do not reach 1 mm/day.

A general and progressive intensification in the moisture sources is found in all seasons for the MC and EC periods under the SSP5-8.5 scenario. As for MC (Fig. 1, middle column) in annual terms it is observed is a slight increase in contributions from southern North Atlantic, western Mediterrenean Sea and Iberian Peninsula; the latter showing the largest increment. In addition, a pronounced intensification of PRPs are found in spring, while a decrease in these is observed in the summer months. The same signal is observed for the contribution from the western Mediterranean region. In autumn, the greatest increase is seen for the western Mediterranean region, varying in the range from 0.3-0.45 mm/day.

In regard to EC (Fig. 1, right column), an amplified version of the changes already described for MC is observed. In this respect there is only the exception for the contribution of the PRPs in the summer

months, which in MC was negative, while for EC it turns out to be positive. It is also worth noting that for EC there is a clear decrease in the contribution of the North Atlantic region located above latitude 40°N and nearby the European coast. This decrease is observed in annual terms, and is particularly intense in spring.

For a better interpretation of the results described in the previous paragraphs, Fig. 2 shows the spatially integrated changes for each source region, in terms of percentage change. This figure shows that, although the most relevant changes in absolute terms (mm day⁻¹) are observed for the PRPs, in percentage terms the expected changes are similar for the three source regions, and even slightly higher in MED and NATL. Quantitatively, the percentage changes obtained are relatively large, exceeding annual increases of 30% for both MED and NATL. In any case—as discussed below—it should be noted that when working with the SSP5-8.5 scenario, these results should be interpreted as the upper bound of what should actually be expected.

Having very long time series has also allowed us to analyse changes in the variability of the results presented. In particular, Supplementary Fig. 5 shows the changes relative to the historical period in the standard deviation of the time series of the contribution of the different sources to precipitation in the Iberian Peninsula. The cited figure shows the percentage changes in the standard deviation of the future periods with respect to the standard deviation of the historical period. Thus, positive values will indicate an increase in variability; while negative values will show a decrease in variability. In general terms, the variability is also expected to increase, particularly for EC, in values close to 15%. There are some exceptions, such as the case of PRP

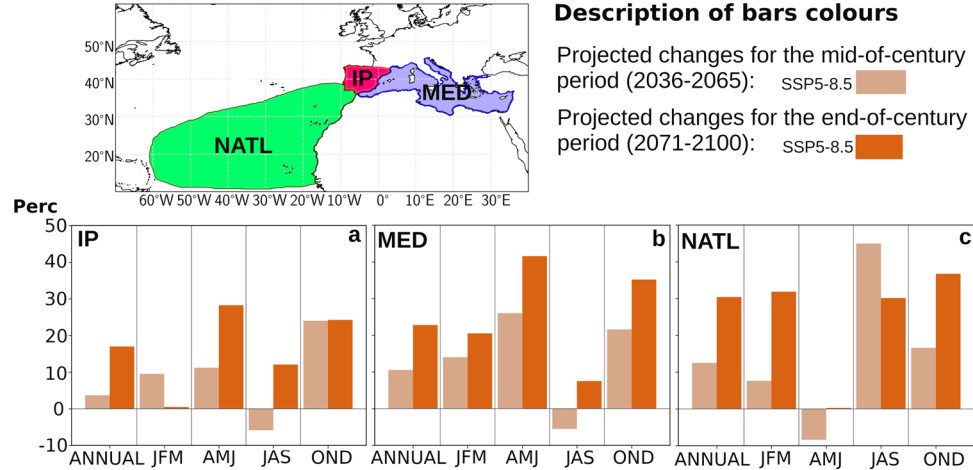

**Fig. 2 | Future changes in the relative contribution (in %) of the moisture sources to the Iberian Peninsula.** Relative changes are shown for (**a**) the pre-cipitation recycling processes (PRPs) in the Iberian Peninsula (IP), (**b**) Mediterranean Sea (MED) and (**c**) North Atlantic Ocean (NATL) over the four seasons as well as in annual terms both for the mid- and end 21st century (2036–2065, MC, and 2071–2100, EC, respectively).

variability, which is expected to decrease slightly in the winter months for both EC and MC.

## Changes in precipitation contribution from the Mediterranean and North Atlantic moisture sources

Beyond the analysis carried out for the Iberian Peninsula, the forward trajectories from the oceanic moisture sources (NATL and MED) were used to evaluate future changes in their precipitation contributions (PCs, total mean amount of precipitation (in mm/day) provided by the sources to the study region[33]) over the surrounding continental areas (see 'Methods' section). Figure 3 (left column) provides regions with P-E > 0 having quantified the moisture balance for the air parcels origi-nating from MED for the historical period. Overall, a remarkable PC is observed in northern and eastern Europe, as well as in regions of North Africa, with contributions of between 3 and 7 mm/day (Fig. 3a). In addition, these PC patterns show that MED provides a similar con-tribution over the continent adjacent to both the north and east of the MED basin in winter and autumn (Fig. 3d, m). In the warm season, the most intense PCs move westward and have a greater effect during spring on Europe and during summer on Africa (Fig. 3g, j). These results agree with those of previous studies[17,18].

The projections of the PCs for MC (Fig. 3, middle column) show results that depend on the season considered. In annual terms, sea-sonal signals tend to counterbalance each other, with a decrease near the Italian Peninsula and a slight increase between the Iberian Penin-sula and the British Isles (Fig. 3b). Also, a reduction of the PCs is observed for the regions bordering MED to the north and this is par-ticularly strong in the winter months (Fig. 3e). There is also an important signal of a reduction of the contribution in central Europe in spring (Fig. 3h) and a certain increase of the contribution over central Europe in autumn (Fig. 3n).

Regarding the projections of PCs for EC (Fig. 3, right column); overall, an amplified version of what is observed for MC is found, with the exception of the summer months, where a clear decrease in the contribution is shown for Central Europe (Fig. 3l). Relevant decreases are also observed throughout the year in the south-eastern European region (Fig. 3c) with a significant increase in North Africa. On the other hand, relevant increases in contribution are observed in Central Eur-ope in the winter and autumn months (Fig. 3f, o), with particularly relevant increases in the summer months (Fig. 3l). In autumn, there is a longitudinal increase in the contribution, especially in North Africa, Fig. 3o).

Figure 4 shows the changes already described for MC and EC in Fig. 3, but in relative percentage terms and spatially integrated over the four most relevant sink regions for MED; the Iberian Peninsula (IP), Western Europe (EUwest), Eastern Europe (EUeast) and North Africa (NAfrica). This figure allows to observe how the most remarkable relative changes are expected over EUwest, where in the winter months the contribution will grow by 40% for EC[34], while in the sum-mer months the contribution will decrease by 40% in the same period. Also notable are the expected changes in NAfrica, which grow in all seasons with an average annual contribution close to 30%. On the other hand, the changes in EUeast—even though they may present significant seasonal values—tend to be temporarily compensated in annual terms. In relation to IP, it is observed how MED will continue to be a main source of moisture for this region in the summer months increasing its influence in values close to +20% for EC, relative to the current values[35]. In the autumn and winter seasons, the role that MED will presumably play in IP is more complex, observing opposite behaviours between MC and EC, which do not allow obtaining clear conclusions in this regard[36].

It is observed a no a priori result concerning the local contribution (particularly associated to the MED, Fig. 4) for regions such as EUwest in AMJ and EUeast in JFM and JAS in which accentuated behaviours are observed for MC with respect to EC or even opposites are observed between both periods. This behaviour would have an explanation of dynamic character, and the explanation can be intuited by considering together the IVT fields (shown in Supplementary Fig. 8) together with the changes in Z500 (Supplementary Fig. 3). Particularly, in the latter, a latitudinal shift of the geopotential height fields is expected to deter-mine different stability conditions for EC and MC over central Europe[37]. The aforementioned changes in stability conditions would decrease the convergence of moisture flux, and thus precipitation; this could be an explanation for the existence of more pronounced local changes in EC and MC at these seasons.

Supplementary Fig. 6 shows a variability analysis analogous to that presented in Supplementary Fig. 5 but for the contribution of precipitation from the MED source to its sinks (IP, EUwest, EUeast and NAfrica). In general, an increase in variability with region-dependent values is again observed. For example, the highest values—close to 25% in annual terms for EC—of variability increase are expected for NAfrica, while for IP they remain close to 15%. The case of EUwest is particularly noteworthy, as it shows an increase of 40 percent for JFM, while a decrease in the summer months is expected for the end of the century.

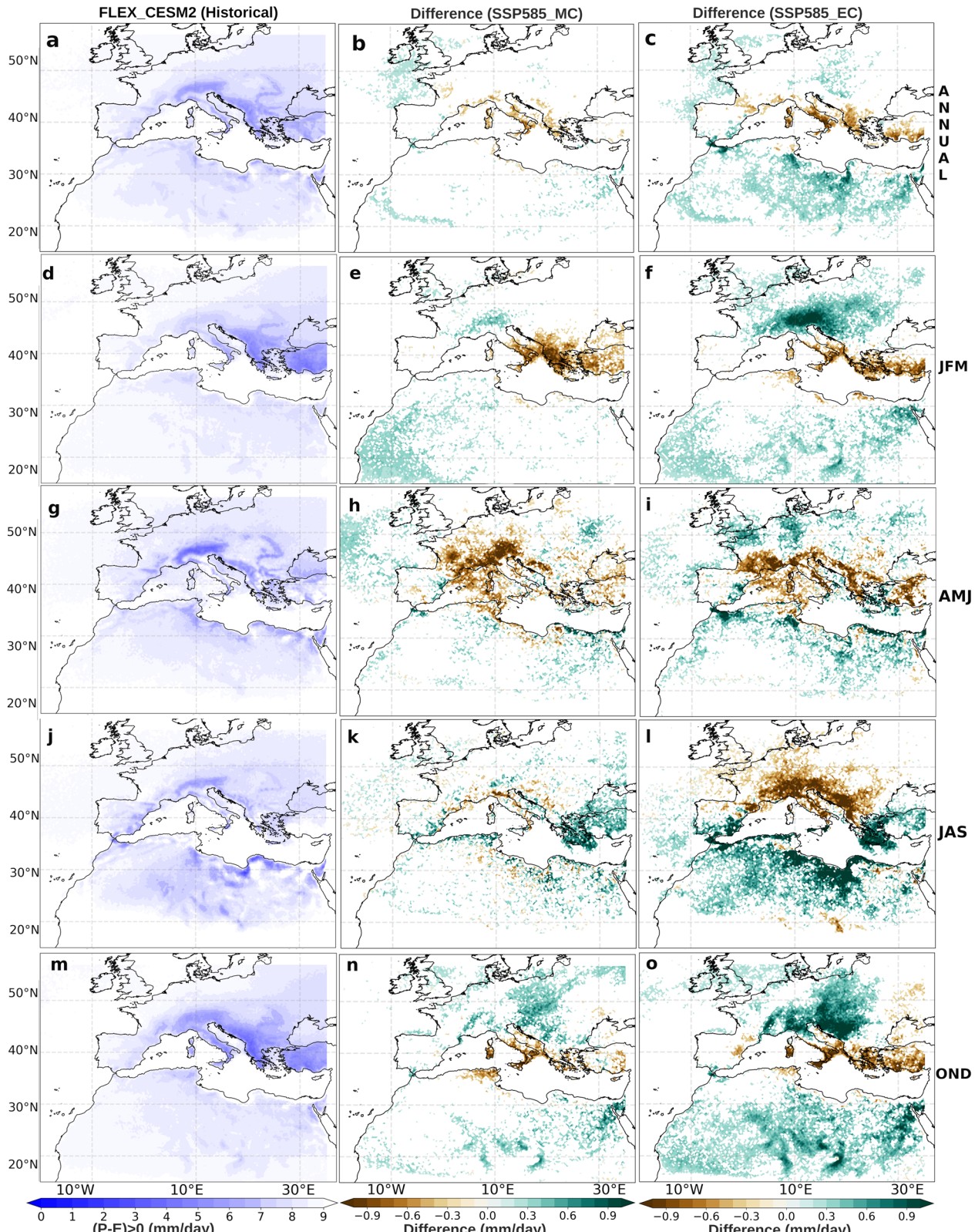

**Fig. 3 | Future changes in the absolute contribution of precipitation (in mm day⁻¹) to moisture sinks from the Mediterranean source.** Moisture sinks fields (P-E > 0) for the Mediterranean Sea source (MED) for the historical reference period (1985–2014, (**a**, **d**, **g**, **j**, **m**)) and differences under the SSP5-8.5 scenario for the mid- and end 21st century (2036–2065, MC (**b**, **e**, **h**, **k**, **n**), and 2071–2100, EC (**c**, **f**, **i**, **l**, **o**), respectively). The fields displayed from top to bottom correspond to annual, winter, spring, summer and autumn periods (ANNUAL, JFM, AMJ, JAS and OND).

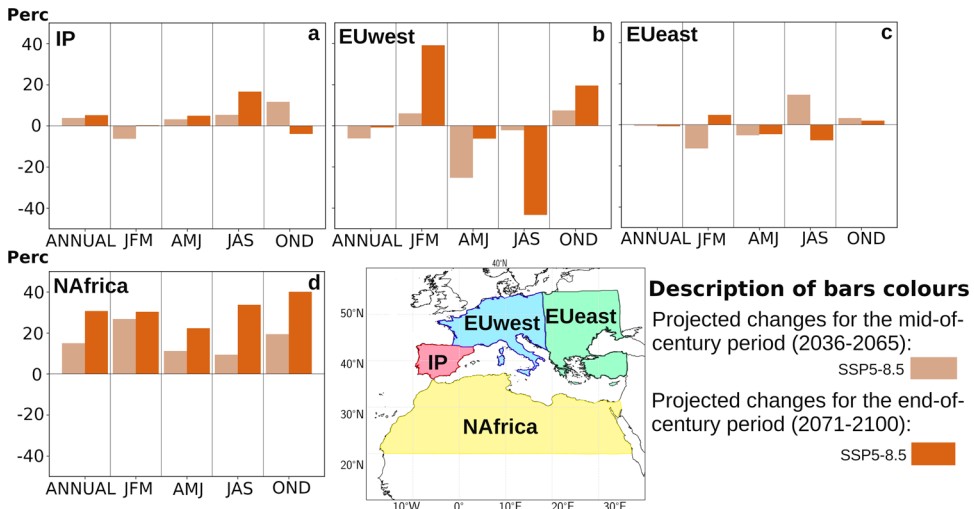

**Fig. 4 | Future changes in the relative contribution of precipitation (in %) to moisture sinks from the Mediterranean source.** Percentage of projected future changes in precipitation contribution over: (**a**) Iberian Peninsula (IP), (**b**) Western Europe (EUwest), (**c**) Eastern Europe (EUeast) and (**d**) North Africa (Nafrica) associated with the Mediterranean Sea (MED) source.

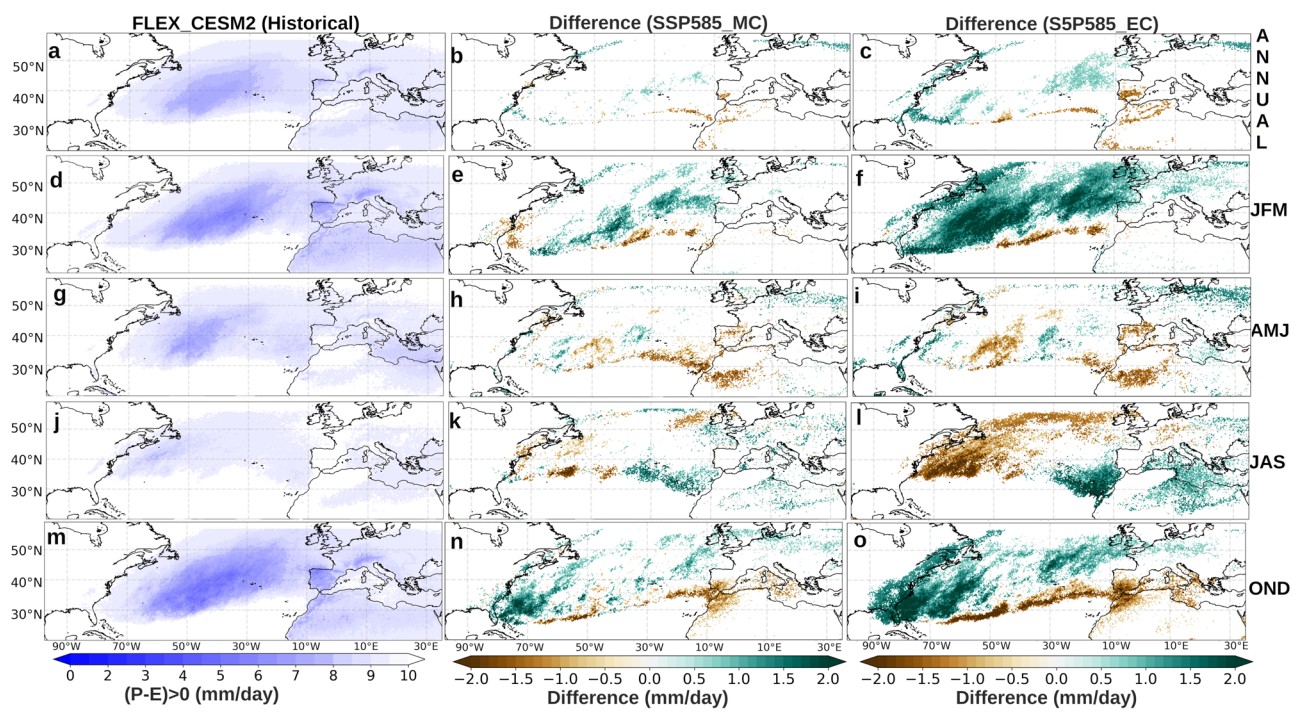

**Fig. 5 | Future changes in the absolute contribution of precipitation (in mm day⁻¹) to moisture sinks from the North Atlantic source.** Moisture sink fields (P-E > 0) for the North Atlantic Ocean (NATL) source for the historical reference period (1985–2014, (**a**, **d**, **g**, **j**, **m**)) and differences under the SSP5-8.5 scenario for the mid- and end 21st century (2036–2065, MC (**b**, **e**, **h**, **k**, **n**) and 2071–2100, EC (**c**, **f**, **i**, **l**, **o**), respectively). The fields displayed from top to bottom correspond to annual, winter, spring, summer and autumn periods (ANNUAL, JFM, AMJ, JAS and OND).

Figure 5 shows results analogous to those presented for Fig. 3, but now for the NATL moisture source. The first column shows how this source of moisture is relevant not only for itself but also for much of Europe[20] and North Africa, particularly in the winter, spring and autumn months and annually (Fig. 5a, d, g, m), as well as for the Mediterranean[18] itself and—although to a lesser extent—for the U.S. East Coast. In the summer months (Fig. 5j), this influence, although persisting, is much more limited, due to the reduction of the baroclinic dynamical systems responsible for most of the oceanic advection.

In relation to the expected changes for MC (Fig. 5, middle column), there is a significant seasonality in these results, which in annual terms tend to cancel each other. For example, a strong increase in the contribution is observed near the North American east coast, together with an intense decrease in the southern IP and Maghreb area for the autumn months[38] (Fig. 5n). This signal is observed—although to a lesser extent—for the winter months (Fig. 5e) and fits with the already predicted northward shift of the PCs[39,40].

Likewise, in the winter months, the contribution along the North Atlantic corridor seems to strengthen with values ranging from 1 to 2 mm/day (Fig. 5e).

Again, for EC (Fig. 5, right column) what is observed is an amplified version of the pattern for MC. In addition to the results already

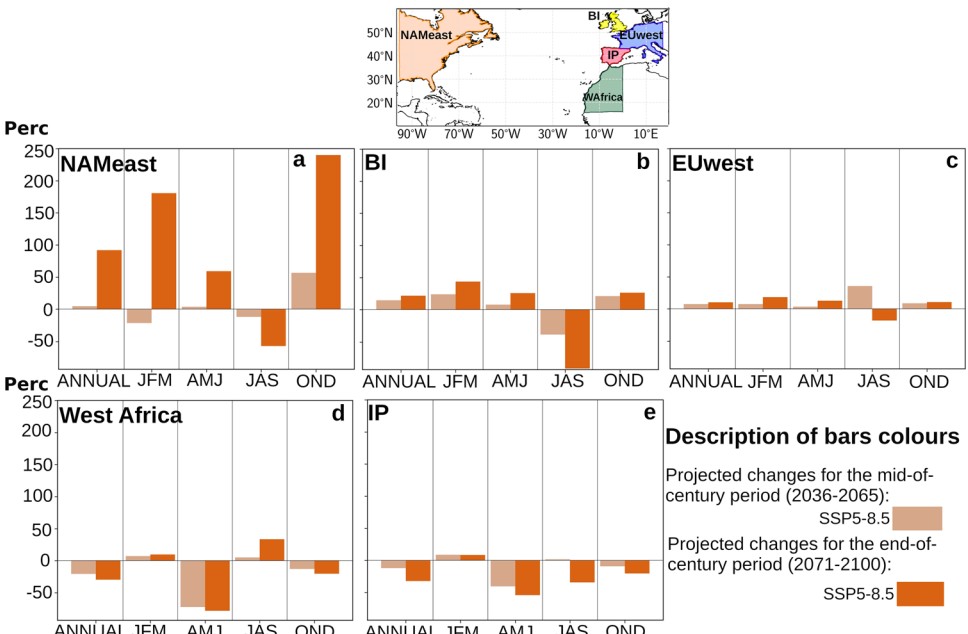

**Fig. 6 | Future changes in the relative contribution of precipitation (in %) to moisture sinks from the North Atlantic source.** Percentage future changes in the precipitation contribution over: (**a**) North American East Coast (NAMeast), (**b**) British Isles (BI), (**c**) European West Coast (EUwest), (**d**) West Africa (WAfrica) and (**e**) Iberian Peninsula (IP) associated with the North Atlantic Ocean (NATL) source.

discussed, it is observed an intense decrease of the contribution in the summer months (>2.0 mm/day) in the northern part of the North American west coast (Fig. 5l). All these results fit with others previously obtained with alternative methodologies, such as those predicting an increase in Atlantic PCs in winter and autumn for south-eastern North America[41] and the generalized decrease in contribution for the African coast[42] or the southern Iberian Peninsula[2,43]. It is also noted that the shift between positive and negative contributions in the North Atlantic —particularly in the winter months—is most likely related to the gradual northward shift of the North Atlantic baroclinic corridor.

Figure 6 shows the observed changes in the contribution of NATL to its main sinks—North American East Coast (NAMeast), British Isles (BI), European West Coast (EUwest), West Africa (WAfrica), and the Iberian Peninsula (IP)—in relative terms. This figure is especially enlightening as it shows, for example, how for EC the contribution on NAMeast increases in annual terms by almost 100%, being the autumn and winter months the largest contributor to this increase. Likewise, the gradual loss of relevance of NATL for WAfrica and IP, as well as for BI in the summer months, is striking. All these results denote a gradual northward shift of the North Atlantic baroclinic corridor, which extends its zones of influence northward, particularly in the extended winter months[1,33,44].

Finally, Supplementary Fig. 7 shows analogous results presented for MED in Supplementary Fig. 6, but for the increased variability in the relative contribution of NATL precipitation over its sources (BI, EUwest, IP, West Africa and NAMeast). Again, with some exceptions at the seasonal level, an increase in variability is expected for all sources except West Africa, particularly for EC. NAMeast stands out at the top, with an increase of 30% in annual terms. In the case of West Africa, a decrease in variability is observed, quantified at −10% in annual terms.

### Synthesis

The hydrological cycle is projected to increase in intensity under climate change[45]. We found a general increase in moisture reaching continental areas in the extratropical North Atlantic and Mediterranean belts. We also observed a significant increase—although to a lesser extent—of the precipitation recycling processes (PRPs) over the Iberian Peninsula (IP) in the winter months. In annual terms, we observed an increase in these PRPs of between 2 and 8%, within the range previously established by Findell et al.[14] although these are doubled for EC. In any case, the projections presented in this article coincide with those that had already identified an increase in the contribution of moisture from oceanic sources—and, to a lesser extent, from the recycling processes—with global warming[15].

The results obtained in relation to the increases in NATL and MED moisture contributions to the IP are consistent with expectations. In particular, they show for the end of the century an increase in the water storage capacity of the atmosphere compatible with Clausius-Clapeyron amplification[46]. This would imply an increased evaporation in the source regions such as the MED and NATL, with an accentuated moisture supply that, at least in part, would end up affecting the IP. Specifically, Supplementary Fig. 9 shows a comparison for North Atlantic Ocean and the Mediterranean Sea of the evaporation field obtained from WRF-ARW outputs between the historical period and MC and EC for the SSP5-8.5 scenario. This figure shows a remarkable increase in evaporation over the entire Mediterranean Sea, particularly for EC. On the other hand, the North Atlantic region shows a clear differentiated pattern between the Northern and Southern sectors, with evaporation decreasing in the former and increasing in the latter; also particularly noticeable for EC. Moreover, they show a slight latitudinal shift from this source region to the EC. Finally, the results show a clear decrease in the contribution of the NATL source in the areas located to the northeast of the North Atlantic near the European shores. These changes may be related to a considerable decrease in evaporation for MC and EC in the aforementioned areas (Supplementary Fig. 9). In addition, the increase of the Z500 field could influence the general anticyclonic circulation, as well as an increase of atmospheric stability situations (Supplementary Fig. 3). This Supplementary Fig. 3 is similar to Supplementary Fig. 9 but considering the Z500 field and the outputs of the CESM2 model. It is highlighted that with this increase of Z500, a greater increase of blocking situations in future periods can be expected mainly in regions such as Europe and the Mediterranean Sea, where the increase is more noticeable. This is

referenced from the results found by Davini and d'Andrea[47] with different CMIP6 climate models. However, there are numerous discrepancies in the results provided by the different models in the work of these authors; as well as considerable biases in the representation of the present climate[47] regarding blocking situations. Thus, it is hasty to take these conclusions into account when determining the climate change signal. Instead, a future evaluation of the behaviour of this model in the Z500 representation becomes necessary.

Higher contributions of the MED moisture source to precipitation are observed in the regions located south of the Mediterranean Sea, both for MC and EC. This contribution will also increase–although to a lesser extent–over the Iberian Peninsula, mainly in the summer season, being more noticeable for EC. There is also observed a decrease in the contribution in Eastern Europe and Western Europe (in summer and spring)[48–50]. This result may be associated with the projected latitudinal shift of storm trajectories in the future climate[40]. In addition, much of this behaviour could be attributed to changes in atmospheric dynamics for both MC and EC. Specifically, Supplementary Fig. 8 shows the changes in the IVT fields for MC and EC with respect to the historical period for the Mediterranean area under the SSP5-8.5 scenario. This figure shows that for both periods changes in the moisture flux patterns are to be expected. Over the Mediterranean region, changes are more noticeable in the eastern region where a weakening of moisture transport towards Eastern Europe is observed. On the other hand, Z500 fields are expected to strengthen in general, with notable values on a regional scale (e.g. Western European and Mediterranean Sea regions). This behaviour could favour a stronger anticyclonic atmospheric circulation in the future, as well as greater stability. For regions with a local strengthening, it could also lead to an eventual increase in the frequency of blocking flows from their moisture sources. This evidence indicates that these changes in atmospheric dynamics may be as or more relevant than the thermodynamic changes that lead to greater availability of moisture in the atmosphere. Further, these results show correspondence with those obtained by Batibeniz et al.[51] over Eastern Europe, showing negative trends for seasonal precipitation trends per year using four reanalyses in summer.

The contribution from the NATL source will increase on the east coast of North America (mainly in winter and autumn), followed by the British Isles (mostly in winter), and gradually decrease in latitude from the northern areas of Western Europe, IP and west coast of Africa. It is notable that a decrease is projected in summer for all the regions analysed. Moreover, these results are consistent with the decrease in moisture transport from the North Atlantic source over parts of the western European regions in winter and summer[51]. These results show that although the NATL source will intensify in the future and provide more moisture to the IP (Fig. 2c), it will not positively contribute to the final amount of precipitation over the IP. Moisture from the NATL source may not necessarily precipitate if the appropriate conditions are absent, such as favourable moisture convergence, forcing for vertical ascent, and instability[52]. This will be a consequence of the poleward shift of the storm tracks and the upward expansion of the midlatitude baroclinic regions[53]. This result can be corroborated with the projections of the total precipitation field according to the CESM2 model for IP, where a general decrease is expected, being more notable at the end of the century[27–29] (Supplementary Fig. 2, analogous to Supplementary Fig. 3 but for the precipitation field). This behaviour in the precipitation contribution from the NATL source to its sink areas agrees with the projections of the poleward movement[1] of the general circulation[33,40]. This displacement will be clearly related to the shift in the trajectories of extratropical cyclones, since these are the main mechanism of this poleward transport of moisture[54]. This potential shift, which has already been observed in recent decades, is also expected to intensify under less optimistic climate change scenarios[55]. Therefore, these regions, mainly Europe, will receive a reduced

contribution from NATL source, which will have an impact on the precipitation regime and a reduction in rainfall, as previously reported for the Mediterranean area[53], especially in IP during winter or autumn and mainly at the EC[7]. This behaviour may be due to the possible extension of stable and dry summer conditions and a decoupling between moisture availability and dynamic forcing[7], but it could also be the product of a circulation that relates to the mean shift of the humidity corridors and the associated atmospheric rivers toward the poles[7,56].

This study makes a significant contribution to the hydrological cycle research because no previous studies based on Lagrangian approaches have considered how climate change will alter the location and importance of moisture source regions or the future moisture transport in the north Atlantic area. These results show that climate change has a large influence on moisture transport around the Atlantic Ocean, which will result in changes in availability of water resources in various regions. These changes could result in water stress, particularly in southern Europe, accentuating long periods of drought and heat waves (very noticeable in summer), generating serious stress on ecosystems and society[57]. As consequences of this warming and the increase in food demand, they can lead to the northward expansion of world agriculture, weakening those of these regions[58]. On the other hand, in winter the events of extreme precipitation in higher latitudes would increase, generating floods for these regions[59] associated with a greater number of atmospheric rivers that reach these latitudes and in turn accentuating winter droughts in southern Europe.

## Methods
### Data employed
The available outputs of the climate model Community Earth System Model Version 2 (CESM2)[60], from Phase 6 of the Coupled Model Intercomparison Project (CMIP6), were dynamically downscaled. The CESM2 data were downloaded from the Earth System Grid Federation (ESGF2) and obtained for the native "gn" grid with a resolution of 0.9 × 1.25 (~1°) presented as an output mesh with 288 × 192 longitude/latitude, 32 vertical levels (top level at 2.25 mb). Specifically, Weather Research and Forecasting model (WRF-ARW) in its version 3.8.1[61] was used to downscaled CESM2 data (see Supplementary Section 1.2). The highest shared socioeconomic pathway (SSP) scenario from the CMIP6 climate projections (SSP5-8.5) has been also used to analyse the different ranges of future forcing pathways to 2100 (see Supplementary Section 1.1). A set of 30-year periods were compared spanning the historical period 1985–2014 and the intervals 2036–2065 (for the mid-century: MC) and 2071–2100 (for the end-century: EC). ERA5[62] reanalysis data were also used to evaluate the results for the historical period. ERA5 is chosen as a reference since this model provides the advantages of high resolution (31 km horizontally and 137 levels vertically) and a large number of assimilated historical observations. In addition, ERA5 significantly improves upon its predecessor, ERA-Interim reanalysis, particularly with respect to precipitation fields both over extratropical regions and tropical oceanic areas. A more detailed description is presented in the Supplementary Information.

### WRF-ARW and FLEXPART-WRF setups
The parameterisations employed in the WRF-ARW configuration were as follows: the WSM6 microphysics scheme[63], Yonsei University planetary boundary layer (PBL) scheme[64], revised MM5 surface layer scheme[65], United Noah Land Surface Model[66], shortwave and longwave RRTMG schemes[67] and the Kain-Fritsch Ensemble cluster scheme[68]. The WRF-ARW outputs had 40 vertical layers from the surface to 50 hPa with a horizontal spacing of 20 km and they covered an area of 115.39–42.02°W and 19.41°S–59.51°N (see Supplementary Fig. 1). For the FLEXPART-WRF[69] configuration, we used Hanna's[70] scheme for turbulence parameterisation with the convection scheme activated. This scheme is based on the boundary layer parameters PBL height,

Monin−Obukhov length, convective velocity scale, roughness length and friction velocity[69]. We assumed skewed rather than Gaussian turbulence in the convective PBL. The FLEXPART-WRF has forty levels and 400 × 777 points, where in the output mesh where the particles are released. The outputs had spatial and temporal resolutions of 20 km and 6 h, respectively.

### Identification of moisture sources and sinks

To estimate the moisture sources and sinks, a Lagrangian methodology was applied to follow the changes in the specific moisture content ($q$) over time ($t$, every 6 h) along the tracks described by each atmospheric particle that the atmosphere was divided into. Therefore, these changes[71] can be calculated by

$$(e - p) = m \left( \frac{dq}{dt} \right) \qquad (1)$$

where $m$ is the mass of the particle, and the difference between $e$ and $p$ considers the increase or decrease in the water vapour ratio along the trajectory. Once the individual trajectories of all particles have been calculated, the total surface freshwater flux in each grid cell can be calculated by summing the contributions of all particles traversing a grid area ($A$) at a given time. The total budget was calculated as follows,

$$(E - P) = \frac{\sum_{k=1}^{N} (e - p)_k}{A} \qquad (2)$$

where $E$ represents evaporation, $P$ is precipitation, and $N$ is the total number of particles over the grid area. For this analysis, the particle trajectories were followed for 10 days, the considered average residence time of water vapour particles in the atmosphere[72–74], and the final computed E-P fields were considered as integrated values during this period.

Moisture particles can be tracked backward (forward) in time from a given region to track their direction and determine their sources (sinks)[71,75]. In a backward experiment, the moisture source of a region is defined as an area in which evaporation dominates over precipitation (i.e. absolute positive values of (E-P), E-P > 0), and in a forward mode projection, air masses with a net loss of moisture are detected to determine areas that are moisture sinks (i.e. areas where precipitation dominates over evaporation, E-P < 0 or P-E > 0).

The moisture field patterns were evaluated using a different dataset, periods and statigraphs (see Supplementary Section 1.4). The post-processing of the results for E-P was carried out with the TRansport Of water Vapor (TROVA) software[76].

### Assessment of projected changes

Future changes were assessed as the difference between the (E-P) fields obtained in 30-year intervals for the MC and EC periods (2036−2065 and 2071−2100, respectively), and the historical reference period 1985−2014.

As the Lagrangian model uses dynamically downscaled CESM2 data (WRF-CESM2), we first conducted simulations using ERA5 data in the historical period (see Supplementary Figs. 10−14). Simulations were then conducted using the FLEXPART-WRFv3.3.2 dispersion model[69] to study moisture changes, and the experiments were forced with WRF-CESM2 outputs every 6 h (herein FLEX-CESM2). According to the distribution of the atmospheric mass, the simulation domain (covering 100°W to 40°E and from 15°S to 57°N, see Supplementary Fig. 1) was homogeneously divided into 2 million air parcels (or particles), which were subsequently advected forward in time for the entire study period. Finally, to obtain the (E-P) fields for comparison, the moisture sources and sinks for the target regions selected in this study (NATL, MED and IP) were identified and computed (Supplementary Section 1).

### Limitations

The results presented in this research show acceptable limitations, mainly related to the nature of the climate models used for WRF-ARW forcing. The considered scenario−SSP5-8.5−assumes a social development based almost exclusively on fossil fuels with a radiative forcing of 8.5 W m$^{-2}$. It is therefore the most pessimistic of those considered in CMIP6, and has a low probability of occurrence. Therefore, the results presented in this research should be understood with caution, assuming that the signals presented in them are likely to be amplified signals of the reality. This is a very common way of proceeding, and is used to detect signals that in other more optimistic scenarios would probably go unnoticed.

It should be noted that the CESM2 model can be considered a warm model, since it projects a warming close to 4 °C by the end of the century[77]. In order to determine the impact that this could have on the results presented in this manuscript, we have proceeded to replicate the simulations presented here using an Ensemble that includes 18 climate models for five years of simulation, and to compare their results with those obtained for CESM2 alone for the same period. The results of this comparison are presented in Supplementary Figs. 15−17. In essence, it has been verified that some differences exist for some seasons of the year. However, these differences are limited. In any case, it should be noted that the results presented in this article may be slightly overestimating the expected changes.

Likewise, the WRF-ARW is also subject to uncertainties derived from certain subjectivity in its configuration. Although the parameterizations used in these simulations−aimed at resolving physical processes occurring at sub-grid scales−have been carefully selected based on the literature, the selection of another set of parameterizations could have slightly modified the achieved results.

Throughout the development of this research, a non-negligible overestimation of the IVT fields carried out by the WRF-ARW model when initialized with ERA5 has been detected. This overestimation affects regions such as the North Atlantic corridor, the Caribbean Sea and, to a lesser extent, the Amazon basin and the west coast of Africa. We have verified that this overestimation of IVT is due to an overestimation in the near-surface wind fields. Likewise, we have also detected a certain tendency of WRF-ERA5 to overestimate E-P values over the Caribbean Sea, the central Atlantic and the Mediterranean Sea. The results could be slightly affected by this circumstance.

Finally, the methodology used by FLEXPART in the Lagrangian dispersion of particles is also not free of uncertainty. Previous studies in the literature have detected an overestimation of evaporation and precipitation values. These fluctuations are mainly due to non-physical processes relevant to the phenomenology[75]. Lagrangian models are also victims of necessary simplifications in their formulation, which can lead to biases. In particular, FLEXPART is known to progressively increase the uncertainty of the air cell trajectories with time[78].

## Data availability

ERA5 reanalysis data can be obtained from https://cds.climate.copernicus.eu/cdsapp#!/dataset/reanalysis-era5-single-levels-monthly-means?tab=form and CESM2 model data from https://esgf-data.dkrz.de/search/cmip6-dkrz/. The WRF-ARW outputs are available upon request to the corresponding author. The request for the data is due to the large volume it occupies, which makes it impossible for it to be stored in an online repository.

## Code availability

Code that supports the findings of this study is available upon request from the corresponding author.

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

## Acknowledgements

This work is supported by the SETESTRELO (PID2021-122314OB-I00) project funded by the Ministerio de Ciencia, Innovación y Universidades, Spain. This publication is part of the ESMORGA I+D+i project (TED2021-129152B-C43), funded by the MICIU/AEI/10.13039/501100011033 and the European Union NextGenerationEU/PRTR. Partial support was also obtained from the Xunta de Galicia under the Project ED431C 2021/44 (Programa de Consolidación e Estructuración de Unidades de Investigación Competitivas (Grupos de Referencia Competitiva) and Consellería de Cultura, Educación e Universidade). José C. Fernández-Alvarez acknowledges the support from the Xunta de Galicia under the grant no. ED481A-2020/193. Albenis Pérez-Alarcón acknowledges a PhD grant from the University of Vigo. J. Eiras-Barca thanks the Defense University Center at the Spanish Naval Academy (CUD-ENM) for the support provided for this research. In addition, this work has been possible thanks to the computing resources and technical support provided by CESGA (Centro de Supercomputación de Galicia) and Red Española de Supercomputación (RES) (AECT-2022-3-0009 and DATA-2021-1-0005).

## Author contributions

L.G. and R.N. designed the study; J.C.F-A. and A.P-A. performed the research; J.C.F.-A., A.P.-A. and S.R.-E. analysed the data; J.C.F.-A., J.E.B. and R.N. wrote the paper; L.G., R.N., J.E.B., J.C.F.-A., S.R.-E. and A.P.-A. review the paper.

## Competing interests

The authors declare no competing interests.
