## [Peer Review File · Nature Communications]

REVIEWER COMMENTS

Reviewer #1 (Remarks to the Author):

Projected changes in atmospheric moisture transport contributions associated with climate warming

Manuscript ID: NCOMMS-22-17469

The study has the potential to become a fully relevant article to the concept of Nature Communications. The topic is potentially interesting given the fact that the study provides important information regarding the potential future of moisture sources and sinks shaping the hydrologic cycle over Iberian Peninsula. The subject is novel and has not been investigated in the future period previously. However, there are some important issues that still need to be clarified and resolved before this paper is suitable for publication in Nature Communications. Therefore, I recommend major revisions. My major comments relate to the data and methodology. My other criticisms are focused on the presentation of some results. I have listed below some suggestions that I hope the authors consider.

Major Comments

1. A known issue in some CMIP6 models is extreme warming. The global temperature anomalies with respect to preindustrial period (1850-1900) shows that CESM2 will reach to 3°C between years 2072-2091 (SSP245) and 4°C between years 2075-2094 (SSP370), 2060-2079 (SSP585) (Seneviratne and Hauser, 2020). Therefore, CESM2 can be counted as one of those warm models. Increasing global warming will increase evaporation from oceanic sources as also mentioned in the manuscript. Therefore, model selection is very important. I invite authors to explain why they chose this model.

Seneviratne, S.I., Hauser, M., 2020. Regional Climate Sensitivity of Climate Extremes in CMIP6 Versus CMIP5 Multimodel Ensembles. *Earths Future* 8. <https://doi.org/10.1029/2019EF001474>

https://github.com/mathause/cmip_warming_levels

2. I think one model is not sufficient to come to a robust understanding regarding the future of sources and sinks of Iberian Peninsula. Therefore, I suggest authors repeat these analyses with at least one more model that doesn't warm as much as CESM2 to be able to give a range for future responses.

3. In my opinion, using five years to define future moisture sources/sinks is not sufficient. It would have been nice to see a longer time period of analysis to be able to see these results more from a climatological perspective.

4. I am surprised that authors have not discussed anything about the limitations of the FLEXPART model and associated uncertainties in the methodology. I understand journal has word limits; however, it is worth mentioning this because the reader would not be aware of the weaknesses of methodology and tracking algorithms.

Minor Comments

1. Pg. 2, line 36, 37, Introduction. The following sentence is a repetition of sentences in lines 33-35. Therefore, I suggest deleting this sentence; "This is expected since evaporation will increase from the world's oceans."

2. Integrated water vapour transport field seems overestimated by WRF in Supplementary figure 2. Do the authors have an idea if this is caused by specific humidity representation of WRF? I suggest authors checking what is causing this difference. I think this will help to evaluate WRF-CESM2 simulation in the future period.

3. Connected with previous comment; Do the authors think WRF simulated precipitation and evaporation is similar to ERA5. I believe FLEX-WRF's performance is directly related with these fields' representation. I believe showing these fields would increase the confidence in using the WRF model.

4. In supplementary figure 3,4,5 you compare E-P to moisture flux divergence. Would it make sense to compare the model's E-P with ERA5's E-P as well?

5. Pg. 5, line 98-101. It is written as “previous studies”, but there is only one reference. Please provide more references supporting this statement or correct the sentence.

6. Pg. 5, line 101. This is very short to explain differences between scenarios.

7. Supplementary figures, eg. Sup. Fig. 8 and 9 should follow Sup. Fig. 7's order to understand how location of sources and sinks change between EC and MC under different scenarios. I believe this will make it easier to follow results, but I will leave the decision to the authors.

8. Pg 6. Section: Changes in moisture sources for the Iberian Peninsula. What does moisture recycling processes (PRPs) mean? Can you please elaborate?

9. Page 8. Lines 153-155. I agree that the precipitation contribution is similar. However, in autumn there is a weird result over Italy. The amount of increase for MC is higher than EC for SSP245 scenario whereas it is the other way around for SSP370. Could that be a plotting mistake? I would mention this opposite bias with one sentence if it is not a plotting mistake.

10. This is one of the main results of this study; “Precipitation associated with the MED will decrease over eastern Europe, while that associated with the NATL will decrease over western Europe and Africa”. Similar trends have been found previously for the historical period). With respect to this, I recommend citing studies showing similar trends for the North Atlantic and the Mediterranean Sea.

- Batibeniz, F., Ashfaq, M., Önoel, B. et al. Identification of major moisture sources across the Mediterranean Basin. *Clim Dyn* 54, 4109–4127 (2020). <https://doi.org/10.1007/s00382-020-05224-3>

Reviewer #2 (Remarks to the Author):

Review of paper: Projected changes in atmospheric moisture transport contributions associated with climate warming by Fernández-Alvarez et al.

This study addresses the question how moisture sources, and the related precipitation, changes in a future climate with increased atmospheric moisture and possible changes in large-scale dynamics. This is investigated for the region of the Iberian peninsula, and the two main moisture sources for this region which are the North Atlantic area, and the Mediterranean Sea. The authors apply a moisture tracking model on the outcomes of a CMIP6 simulations of one model, based on present climate, and future climate under different scenario's (SSP2-4.5, SSP3-7.0 and SSP5-8.5) and at different moments in time (EC and MC) of 5-year moisture tracking simulations.

The authors find an annual increase of moisture recycling over the Iberian Peninsula, although varying per season. Furthermore, from forward tracking of the moisture from the Mediterranean and North Atlantic basin it is found that those sources will increase, but impact different regions differently.

I have a few major concerns regarding the manuscript in its current form that I will explain in more detail below. In my opinion these concerns should be thoroughly addressed before publication is possible. My largest concern relates to the only 5-year simulations of the moisture sources for present and future climate, on which the conclusions are based. Given the internal variability of our climate, these simulations should be extended before solid conclusions can be given (see my major concern 1). Furthermore, I have listed some more specific comments afterwards. Please note that this is not an extensive list of specific comments, rather comments that I noted down while reading through the manuscript.

Major concerns

1) Methodology

My largest concern on this work is that the results of this study are based on only 5-years of simulations for current and future climate. This is a very short period to draw conclusions from, taking into account the internal variability of our climate. By only analysing five years, it could happen that your results are biased to one or two dry or wet years being present in your data. I argue it is needed to extend the simulations of moisture sources before solid conclusions can be drawn. In terms of comparing climatologies of present and future climate, it's common to work with 30-year simulations, so the mean sources can be compared between two climates. By showing distributions of moisture sources, or performing significant tests between present and future climate, one can then address if the changes found in a future climate are actually significantly different. Now increases or decreases in sources can also happen from large anomalies in the 5-year data, instead of being an actual difference between present and future climate.

While the limitation regarding using one climate model is mentioned under Limitations of this study, the shortness of the simulations being 5 year present and future climate is not mentioned in the manuscript at all, while I think it is a major issue regarding the methodology (and the related conclusions). I realise that it is a major effort to lengthen the simulations by 6-fold (for each scenario and period) to get robust results from 30-years of simulations, but it is needed to prove climate signal over internal variability.

2) Novelty of results and applying moisture tracking to future climate simulations

The authors mention that they are the first to assess moisture sources using a Lagrangian tracking method in a future climate, on which I agree. However, similar studies have been performed with moisture tracking tools which use an Eulerian moisture tracking approach, and also applied this on future climate simulations (Findell et al. 2019 & Benedict, 2021). While the methodology to perform the tracking is different in those studies, the output product of the moisture source is similar, as is shown in earlier comparison studies of Eulerian and Lagrangian moisture tracking techniques (Van der Ent, 2013). The results from these earlier studies are in line with the results presented in this study as I explain below. Findell et al (2019) studied global moisture recycling in a future climate and concluded that oceanic moisture sources will become more important in the future. This is a similar conclusion as drawn in this study. The same conclusion, that moisture sources will become more important in a future climate was given by Benedict et al (2021), although Benedict et al only focused on the Mississippi basin. Specifically on the Iberian Peninsula, Findell et al. (2019) reports an increase in precipitation recycling ratio towards the future, which is also concluded in the findings of this paper.

Thus, although the exact technique to perform the tracking is not applied before on future simulations, the results drawn in this study are not necessarily surprising, but confirm findings from earlier studies as is also stated at multiple locations within the manuscript. Therefore I wonder if the results presented here show enough novelty to be published in this high-impact journal.

Of course this works specifically focuses on the Iberian Peninsula, and the closely related surrounding atmospheric moisture sources. However, this is now not well expressed in the title of the manuscript. Therefore, I would suggest to include the region in the title of the manuscript, as the title now suggests a global study, which is not true. For example the title could be changed to: Projected changes in moisture sources and sinks of the Iberian Peninsula associated with climate warming.

2) Presentation and interpretation of results and discussion

This study is analysing the changes in moisture sources towards the Iberian Peninsula, and the changes in precipitation from the source of the North Atlantic and Mediterranean. For me, the starting point of such analyses is to first look at the simulated precipitation itself. By doing so, the interpretation of the absolute moisture sources, increasing or decreasing, can be put in the perspective of decreasing or increasing precipitation in a changing climate. Right now, it remains unclear to the reader how the precipitation itself is changing towards the future, as only the moisture source results are discussed. I would suggest to start the result section by showing the changes in precipitation over the Iberian Peninsula (and the other land regions affected by the Mediterranean and North Atlantic) from the CESM model.

The term precipitation contribution is not a term which is familiar to the wider hydro meteorological community that can be reached with this article. If such a relatively unfamiliar term is introduced it requires a good explanation to be able to have a good interpretation of the results of this variable. Now, the results feel like an enumeration of findings on increases and decreases in the precipitation contribution, but it is not clear what the implications are of these findings. Related to my previous comment on showing absolute precipitation changes, this will also benefit the interpretation on the precipitation contribution (PC) from oceanic moisture source results. Now it remains unclear if the precipitation contribution change because there is a relative increase/decrease of precipitation contribution because there is an actual larger relative contribution from the oceanic source, or because the total precipitation in the region is changing. Or another reason is that the dynamics are changing shifting the precipitation sinks (from the oceanic sources) to different regions. It would be very insightful to get a better interpretation of the precipitation contribution results.

The discussion as presented in the paper at the moment repeats many of the results presented before, and here and there, connects those results with the existing literature. To me, it feels as an almost complete repetition of the results with some extra interpretation. This makes the discussion very lengthy and not to the point. I would suggest to either shorten the summary of the results in this discussion section and emphasize the real discussion part. Or to move the discussion of the results directly to the result section and only provide a short summary and outlook in the last section which you could call synthesis.

In general, I would like to see more of an outlook of the implications of your results as now in the discussion mostly numbers and connection with existing literature is given, while I would like to hear more about what these results in the field of precipitation, land-atmosphere interactions etc. This is what happens a little bit at the end of the discussion section, but is not highlighted very clearly because of the lengthy section.

Other more specific comments

Line 22-23: 'Precipitation associated with the MED ..  this is an unclear sentence

Line 36-37: 'This is expected since evaporation will increase ' here you repeat almost exactly on what was said two sentences before (line 34 '., an increase in mean annual evaporation is expected, '. Same holds for the sentence on IVT.

Line 47-48: Changes in moisture recycling in the Iberian Peninsula. Are the numbers you give here based on the research of this paper? If so, I would mention it only later in the conclusion. If not, it needs a reference, and then you can compare it to the numbers you get in this study.

Line 57-58: Can you supply some more motivation why you focus on the Iberian Peninsula? And why the MED and NATL are defined as they are? (for the MED region it is obvious, but the reason for selecting such a specific NATL region is not clear to me)

Lines 65-66: You should mention in the main text what is the time period that you have done the analysis for and how that influences your results (see major point on methodology)

Line 65: global climate models  should be global climate model as you only use one model, have you checked if this model gives a good representation of precipitation and circulation around the Iberian Peninsula? That could be a verification for using the CESM model. Now it is unclear why you have picked that specific model to do the analyses for.

Line 99-101: I think it is also good to link to research about the storm track here, as that is basically what you are analysing, moisture transport within extra-tropical cyclones.

Line 202-203: Here you mention a 'decrease in recycling ratios' while the conclusion was just that there was an increase in recycling ratio over the Iberian Peninsula (line 189-190). This seems very contradictory to me, can you please comment and also verify in the manuscript

Supplementary Figures 10 & 11, and other similar figures: Can you combine the information in those figures by giving percentages of source contributions per region in barplots, given the different scenarios and timesteps. In this way it will be much easier to compare the different simulations.

It is very hard to give good comments on the figures if the caption are not directly with the figure (now you have to count the number of figures and then relate it to the caption). Besides, I think some figures in the main manuscript are actually figure which belong to the supplementary figures (as the supplementary material does not contain any figure), which is very confusing and makes it hard to provide good feedback on the figures.

The splitting up in seasons regarding the figures with the barplots (which I think do present the results in a concise manner) is to me a bit illogic as I use to think in seasons: DJF, MAM, JJA, SON. Is there a reason for the authors to have the seasons like this, and could you support this?

References

Benedict, I., Van Heerwaarden, C. C., Van Der Ent, R. J., Weerts, A. H., & Hazeleger, W. (2020). Decline in terrestrial moisture sources of the mississippi river basin in a future climate. *Journal of Hydrometeorology*, 21(2), 299-316.

Findell, K. L., P. W. Keys, R. J. van der Ent, B. R. Lintner, A. Berg, and J. P. Krasting, 2019: Rising temperatures increase importance of oceanic evaporation as a source for continental precipitation. *J. Climate*, 32, 7713–7726

van der Ent, R. J., Tuinenburg, O. A., Knoche, H.-R., Kunstmann, H., and Savenije, H. H. G.: Should we use a simple or complex model for moisture recycling and atmospheric moisture tracking?, *Hydrol. Earth Syst. Sci.*, 17, 4869–4884, <https://doi.org/10.5194/hess-17-4869-2013>, 2013

Projected changes in atmospheric moisture transport contributions associated with climate warming in the North Atlantic region.

Proposal for publication in Nature: Communications.

ROUND 1 (Jan/Feb 2023)

5

REVIEWER 1

We thank the reviewer for his/her positive evaluation of our manuscript and his/her questions and suggestions which helped to improve the manuscript substantially.

10

The reviewer's main request, related to the (short) simulation period used, has been addressed. The latest version of the manuscript incorporates results obtained with 30-year simulations, instead of 5 years. This improvement has required a large computational effort on the part of the authors, but it was certainly a necessary one. The latest version of the manuscript also includes an analysis quantifying the impact of the use of CESM2 models alone.

15

In this document we will strive to respond to the rest of the reviewer's questions to the best of our ability. The reviewer will find that in addition to the answers to his questions contained in this document, we have made major changes and improvements to the text of the manuscript.

Please, find attached below the one-by-one reply to the specific comments.

Fernández-Álvarez et al.

1. Major Comments

20

1. A known issue in some CMIP6 models is extreme warming. The global temperature anomalies with respect to preindustrial period (1850-1900) shows that CESM2 will reach to 3°C between years 2072-2091 (SSP245) and 4°C between years 2075-2094 (SSP370), 2060-2079 (SSP585) (Seneviratne and Hauser, 2020). Therefore, CESM2 can be counted as one of those warm models. Increasing global warming will increase evaporation from oceanic sources as also mentioned in the manuscript. Therefore, model selection is very important. I invite authors to explain why they chose this model.

25

We do understand the reviewer's concern. The selection of the CESM2 to carry out this research was based mainly on two criteria:

30

- i. **The CESM2 has been previously evaluated for the variables involved in moisture transport.** Specifically, CESM2 data has been evaluated for representing jet streams and storm tracks, Northern Hemisphere (NH) stationary waves, global divergent circulation, annular modes, the North Atlantic Oscillation and NH winter blocking (Simpson et al., 2020). CESM2 ranks within the top 10% of CMIP class models with respect to many of these features (Simpson et al., 2020). In addition, precipitation prediction in CESM2-based subseasonal forecast systems has been shown to be similar to the prediction of the NOAA CFSv2 model, and slightly lower than the prediction of the ECMWF model (Richter et al., 2022). Besides, the NAO —critical to our region of interest— structure in winter and summer is relatively well represented in CESM2 with some minor biases that are quite similar to the rest of the CMIP6 climate models (Simpson et al., 2020). This implies an adequate representation of the associated precipitation anomalies over the Mediterranean (Blade et al., 2012).

40

On the other hand, CESM2 has a remarkable representation of the velocity potential of the upper troposphere in both summer and winter. This element is closely related to tropical precipitation and represents a significant forcing of extratropical standing waves (Simpson et al., 2020). Moreover, it presents improvements in rainfall in regions of great global interest such as the Indian Ocean, East Asia, the tropical Atlantic and the Amazon (Simpson et al., 2020). Finally, CESM2 has been used to study the sea surface temperature effect increase on future changes in Atmospheric Rivers, which is an important mechanism in moisture transport from tropical regions to mid and high latitudes (Gimeno et al., 2016, McClenny et al., 2020, 2021).

45

- ii. **CESM2 is a very convenient model for initializing WRF simulations.** In this regard, it should be emphasized that not only are all the variables needed to initiate WRF-ARW provided directly by CESM2, but also the natural resolution at which these variables are provided is very good. While other models provide natural resolutions of 250 km, CESM2 works with a resolution of 100 km, which has been very useful in previous related work such as, for example, for the simulation of precipitation over the US West Coast (Rahimi et al, 2021, 2022). If we had had to perform interpolations or even had to obtain certain necessary variables from others it would have added uncertainty to the results and would have multiplied the computational effort.

50

A discussion about this point has been included in the Supplementary Material and the following paragraph has been included in the limitations of the manuscript:

Limitations

(...)

It should be noted that the CESM2 model can be considered a warm model, since it projects a warming close to 4°C by the end of the century (Seneviratne and Hauser, 2020). In order to determine the impact that this could have on the results presented in this manuscript, we have proceeded to replicate the simulations presented here using an Ensemble that includes 18 climate models for five years of simulation, and to compare their results with those obtained for CESM2 alone for the same period. The results of this comparison are presented in section 1.5 (Sup. Fig. 10-12) of the supplementary material. In essence, it has been verified that some differences exist for some seasons of the year. However, these differences are limited. In any case, it should be noted that the results presented in this article may be slightly overestimating the expected changes.

(...)

More details about this comparison with Ensembles has been included in the subsequent responses to the reviewer.

2. I think one model is not sufficient to come to a robust understanding regarding the future of sources and sinks of Iberian Peninsula. Therefore, I suggest authors repeat these analyses with at least one more model that doesn't warm as much as CESM2 to be able to give a range for future responses.

It is true that CESM2 is considered a warm model, and this is mainly because it reaches +4°C in the period 2060-2079 when compared to the historical period (Seneviratne and Hauser, 2020). We understand that this may be of concern to the reviewer, and that proof is requested that this does not affect the reliability of the results obtained in this analysis.

Therefore, we have repeated the simulations making use of the forcing provided by the global database CMIP6 **bias-corrected**¹. This data provides an ensemble of 18 models included in the CMIP6 which has been bias-corrected with the outputs of ERA5. The correction was performed with nonlinear techniques applied on the variance, at a spatial resolution of 1.25° and a temporal resolution of 6 hours. More details on this can be found in Xu et al. (2021). Let us call these outputs WRF-ENS, to differentiate them correctly from the original CESM2 simulations (WRF-CESM2).

Figure Rev1.1 shows a comparison between the quantitative increment in the moisture source contributions for the Iberian peninsula with WRF-ENS and WRF-CESM2. Mediterranean sources (MED), North Atlantic sources (NATL) and moisture recycling for the Iberian Peninsula (IP) are included. This figure shows the comparison of future changes in the moisture contribution for the Iberian Peninsula considering the simulations for WRF-CESM2 and WRF-ENS. In this case, the areas corresponding to NATL, IP and MED are used (see masks Figure 1). In general, very similar behavior is observed for the two moisture sources and the moisture recycling processes associated with IP. It is true that some differences are observed in some seasons such as spring and autumn in the moisture contribution in the future climate associated with NATL, showing inverse behaviors between WRF-CESM2 and WRF-ENS.

Figure Rev1.2 shows the comparison of future changes in the contribution to precipitation associated with the Mediterranean source. Similar to the analysis for the IP region, the percentage values for WRF-CESM2 and WRF-ENS tend to be very similar, although differences are also seen for the contribution to IP in the summer season, in the case of Western Europe in winter, slight differences in autumn for Eastern Europe and North Africa in spring.

Finally, a similar analysis for MED is presented for the NATL source in Figure Rev1.3. In this case it is observed that WRF-CESM2 for the eastern region of North America underestimates the percentage values mainly in autumn, where the increase is very noticeable for WRF-ENS. In the rest of the areas studied, the results show similar percentage changes, except for the West Africa and IP regions in spring.

Therefore, the presented comparison between WRF-CESM2 and WRF-ENS for 5 years shows that the projected changes for moisture sources and sinks do not depend considerably on the observed level of warming by the end of the century for the CESM2 model, if the methodology is applied as proposed in the article. However, it should be noted that there are some point differences in the 5-year comparison, which should be smoothed out over a longer period. Therefore, the 30-year increment for the simulations (considering SSP5-8.5) will show robust climate change results for moisture transport in the North Atlantic, even if only one model was used. In future research we intend to extend the investigation with the simulations for 30 years using the model ensemble and other models of our own.

¹Please note that this comparative analysis has been carried out for the original 5-year period used in this analysis, rather than for the new 30-year period. To have used 30 years in the comparison would have required more than 5 months of simulations on our part, making this review process unfeasible.

100 **Figure Rev1.1 (NOT INCLUDED IN THE MANUSCRIPT)**

Figure Rev1.2 (NOT INCLUDED IN THE MANUSCRIPT)

Figure Rev1.3 (NOT INCLUDED IN THE MANUSCRIPT)

105 **3. In my opinion, using five years to define future moisture sources/sinks is not sufficient. It would have been nice to see a longer time period of analysis to be able to see these results more from a climatological perspective.**

Indeed, as we have already stated, the current version of the manuscript analyzes results for 30 years of simulations over the following periods:

- 110 ■ Historical period: 1985-2014.
- Mid-Century (MC): 2036-2065.
- End-Century (EC): 2071-2100.

The computational effort required to increase the simulation periods has been considerable, but it was certainly necessary.

115 **4. I am surprised that authors have not discussed anything about the limitations of the FLEXPART model and associated uncertainties in the methodology. I understand journal has word limits; however, it is worth mentioning this because the reader would not be aware of the weaknesses of methodology and tracking algorithms.**

Certainly, the fact that the journal has a word limit, and that these limitations have been widely discussed in many previous works, have made us initially decide to avoid such discussion.

120 Nevertheless, we consider that the reviewer's proposal is appropriate, so we have proceeded to add the following section to the manuscript:

Limitations

The results presented in this research show acceptable limitations, mainly related to the nature of the climate models used for WRF-ARW forcing. The considered scenario —SSP5-8.5— assumes a social development based almost exclusively on fossil fuels with a radiative forcing of 8.5 W m^{-2} . It is therefore the most pessimistic of those considered in CMIP6, and has a low probability of occurrence. Therefore, the results presented in this research should be understood with caution, assuming that the signals presented in them are likely to be amplified signals of the reality. This is a very common way of proceeding, and is used to detect signals that in other more optimistic scenarios would probably go unnoticed.

It should be noted that the CESM2 model can be considered a warm model, since it projects a warming close to 4°C by the end of the century (Seneviratne and Hauser, 2020). In order to determine the impact that this could have on the results presented in this manuscript, we have proceeded to replicate the simulations presented here using an Ensemble that includes 18 climate models for five years of simulation, and to compare their results with those obtained for CESM2 alone for the same period. The results of this comparison are presented in Sup. Fig. 10-12. In essence, it has been verified that some differences exist for some seasons of the year. However, these differences are limited. In any case, it should be noted that the results presented in this article may be slightly overestimating the expected changes.

Likewise, the WRF-ARW is also subject to uncertainties derived from certain subjectivity in its configuration. Although the parameterizations used in these simulations —aimed at resolving physical processes occurring at sub-grid scales— have been carefully selected based on the literature, the selection of another set of parameterizations could have slightly modified the achieved results.

Throughout the development of this research, a non-negligible overestimation of the IVT fields carried out by the WRF model when initialized with ERA5 has been detected. This overestimation affects regions such as the North Atlantic corridor, the Caribbean Sea and, to a lesser extent, the Amazon basin and the west coast of Africa. We have verified that this overestimation of IVT is due to an overestimation in the near-surface wind fields. Likewise, we have also detected a certain tendency of WRF-ERA5 to overestimate E-P values over the Caribbean Sea, the central Atlantic and the Mediterranean Sea. The results could be slightly affected by this circumstance.

Finally, the methodology used by FLEXPART in the Lagrangian dispersion of particles is also not free of uncertainty. Previous studies in the literature have detected an overestimation of evaporation and precipitation values. These fluctuations are mainly due to non-physical processes relevant to the phenomenology (Stohl et al., 2004). Lagrangian models are also victims of necessary simplifications in their formulation, which can lead to biases. In particular, FLEXPART is known to progressively increase the uncertainty of the air cell trajectories with time (Stohl, 1998).

Stohl, A. & James, P. A Lagrangian analysis of the atmospheric branch of the global water cycle. Part I: Method description, validation, and demonstration for the August 2002 flooding in central Europe. *J. Hydrometeorol.* 5, 656-678 (2004).

Stohl, A. Computation, accuracy and applications of trajectories— a review and bibliography. *Atmos. Environ.* 32, 947–966 (1998).

5. Integrated water vapour transport field seems overestimated by WRF in Supplementary figure 2. Do the authors have an idea if this is caused by specific humidity representation of WRF? I suggest authors checking what is causing this difference. I think this will help to evaluate WRF-CESM2 simulation in the future period.

Indeed, Figure 2 of the supplementary material shows a non-negligible overestimation of IVT by WRF with respect to ERA5. This overestimation is mainly centered in the North Atlantic corridor, both in the westerlies and in the trade wind regions. This overestimation is particularly noticeable in regions where there is an increase in the magnitude of IVT (Rahimi et al., 2022).

As the reviewer is aware, IVT is a derived and vertically integrated variable. Biases in this variable can occur either by bias in the specific humidity, or by BIAS in the wind fields; at any vertical level. Since the vertical levels that weigh the most in the integration are the lower levels, we have proceeded to evaluate both the specific humidity at the 850 hPa level (q850, Figure Rev1.4) and the wind at 10 m from the surface (Figure Rev1.5). The comparison was made between WRF_ERA5 and ERA5 over the period 1985-2014.

From the observation of both images it can be deduced the tendency of WRF to underestimate the q850 fields, while overestimating the near-surface wind values. It is this overestimation of the wind that gives rise to this overestimation of IVT that is detectable in Figure 2 of the supplementary material to the original version of the manuscript.

We have included a brief discussion in the "limitations" section.

Figure Rev1.4 (NOT INCLUDED IN THE MANUSCRIPT)

Figure Rev1.5 (NOT INCLUDED IN THE MANUSCRIPT)

6. Connected with previous comment; Do the authors think WRF simulated precipitation and evaporation is similar to ERA5. I believe FLEX-WRF's performance is directly related with these fields' representation. I believe showing these fields would increase the confidence in using the WRF model.

180 Similar to the analysis of the previous question, Figure Rev1.6 compares the mean precipitation field for the entire North Atlantic region. It is observed that WRF-ERA5 overestimates precipitation values relative to ERA5 in regions of the Caribbean Sea and east coast of the United States, being more noticeable in the tropical zone. However, there is some underestimation over the Central and Eastern Atlantic and seasons such as winter and autumn over Western Europe.

185 In a more detailed analysis of the precipitation field for WRF-ERA5 but focused on Europe (Figure Rev1.7), it is shown that WRF-ERA5 better represents the precipitation pattern in regions with complex orography (e.g., mountainous areas of the Iberian Peninsula) and southern Europe (Fernandez-Alvarez et al., 2022) compared to the ERA5 reanalysis and the MSWEPv2 database (Beck et al., 2019). This demonstrates an advantage of using WRF to simulate meteorological variables in mountainous regions. This allows us to have confidence in the WRF simulations and believe that they can be used to study future changes in the precipitation contribution from the Mediterranean Sea source.

190 As for evaporation (Figure Rev1.8), WRF-ERA5 reproduces the pattern with respect to ERA5, although it generally overestimates the reanalysis pattern. The differences are mainly noted in the regions of maximum evaporation near the east coast of the USA and the Caribbean Sea with slightly higher values and underestimates over the eastern Atlantic. It should be noted that daily precipitation and evaporation are variables calculated with post-processing from WRF model outputs. This implies errors in the calculation associated with the necessary approximations to be used in the determination of these variables; that is, they are not direct WRF outputs.

195 As a summary, the comparison of the moisture flux (E-P, Figure Rev1.9) for the analyzed period is presented. It is observed that the E-P pattern is similar to that obtained for ERA5, correctly representing the main source and sink regions of North Atlantic moisture. However, as mentioned above, WRF-ERA5 shows the same tendency to overestimate the values, in this case of E-P, over the Caribbean Sea, tropical zone and Central Atlantic. It is noteworthy that for winter and autumn it underestimates over the region above 40N. Finally, over the European region it shows an acceptable behavior, but with areas where it overestimates and others where it underestimates. In addition, it is observed that it always overestimates the E-P field over the Mediterranean Sea.

200 The reviewer can check in the previous question that this has already been stated in the "limitations" section.

Figure Rev1.6 (NOT INCLUDED IN THE MANUSCRIPT)

205

Figure Rev1.7 (NOT INCLUDED IN THE MANUSCRIPT)

Figure Rev1.8 (NOT INCLUDED IN THE MANUSCRIPT)

Figure Rev1.9 (NOT INCLUDED IN THE MANUSCRIPT)

7. In supplementary figure 3,4,5 you compare E-P to moisture flux divergence. Would it make sense to compare the model's E-P with ERA5's E-P as well?

Not really. The point is that the $(E - P)_{[1-10]}$ derived from the Lagrangian approach considering 10 days as the residence time of water vapor in the atmosphere (Numaguti, 1999; van de Ent and Tuinenburg, 2017) is not comparable to the E-P calculated by Eulerian approximation. Note that the E-P representation of the reanalysis is simply the freshwater flow that would correspond to the Lagrangian E-P 6h before or after the date for backward or forward scattering.

In short, they could be compared but the comparison would not be thorough, and would induce more error than it would help.

8. Pg. 5, line 101. This is very short to explain differences between scenarios.

The new version of the manuscript includes 30-year simulations. Since each of these simulations requires a computation time of 4 months, it is impossible for us to provide a comparison between scenarios with this time scope. We have preferred to focus our study on the original scenario. We kindly ask the reviewer to understand this.

All references to other scenarios have been removed from the new version of the manuscript.

9. Supplementary figures, eg. Sup. Fig. 8 and 9 should follow Sup. Fig. 7's order to understand how location of sources and sinks change between EC and MC under different scenarios. I believe this will make it easier to follow results, but I will leave the decision to the authors.

The figures in the supplementary material have been rearranged. We ask the reviewer to note that the use of the SSP5-8.5 scenario only —due to the extension of the simulations to 30 years— has altered the original nature of the figures.

10. Pg 6. Section: Changes in moisture sources for the Iberian Peninsula. What does moisture recycling processes (PRPs) mean? Can you please elaborate?

Precipitation recycling processes (PRPs) are simply defined as processes that trigger the portion of precipitation whose origin lies in evaporation over the proper region (Brubaker et al., 1993; Van der Ent et al. 2010).

Indeed, the original version of the manuscript refers to *moisture* recycling processes; this gives rise to confusion. To avoid this confusion, the manuscript has been modified as follows:

(...) predominant influences from the precipitation recycling processes (PRPs) — processes that trigger the portion of precipitation whose origin lies in evaporation over the proper region (van der Ent et al., 2010)— and the MED in summer and spring. Annually, the PRPs provide the greatest contributions.

11. Page 8. Lines 153-155. I agree that the precipitation contribution is similar. However, in autumn there is a weird result over Italy. The amount of increase for MC is higher than EC for SSP245 scenario whereas it is the other way around for SSP370. Could that be a plotting mistake? I would mention this opposite bias with one sentence if it is not a plotting mistake.

With the extension of the simulations to 30 years, and the unique use of the SSP5-8.5 scenario, the discussion of this strange result on Italy is no longer relevant. In any case, we do not believe that this was a plotting mistake, but rather a spurious signal caused by the use of an excessively short simulation period.

The results for SSP5-8.5 and 30 years of simulation are consistent over that region; with similar biases for MC and EC, more intense in the latter.

250 **12. This is one of the main results of this study; “Precipitation associated with the MED will decrease over eastern Europe, while that associated with the NATL will decrease over western Europe and Africa”. Similar trends have been found previously for the historical period). With respect to this, I recommend citing studies showing similar trends for the North Atlantic and the Mediterranean Sea.**

Following the reviewer’s indications, the following ideas have been added to the Summary section:

(...)

255 Higher contributions of the MED moisture source to precipitation are observed in the regions located south of the Mediterranean Sea, both for MC and EC. This contribution will also increase —although to a lesser extent— over the Iberian Peninsula, mainly in the summer season, being more noticeable for EC. There is also observed a decrease in the contribution in Eastern Europe and Western Europe (in summer and spring) (Vautard et al., 2014; Forzieri et al., 2014; Schleussner et al., 2016). This result may be associated with the projected latitudinal shift of storm trajectories in the future climate (Woolings et al., 2012). Further, these results show correspondence with those obtained by Batibenz et al. over Eastern Europe, showing negative trends for seasonal precipitation trends per year using four reanalysis in summer.

260 The contribution from the NATL source will increase on the east coast of North America (mainly in winter and autumn), followed by the British Isles (mostly in winter) and decreases as you descend in latitude from the northern areas of Western Europe, IP and west coast of Africa. It is notable that a decrease is projected in summer for all the regions analyzed. Moreover, these results are consistent with the decrease in moisture transport from the North Atlantic source over parts of the western European regions in winter and summer reported by Batibenz et al. (2020).

265 (...)

Projected changes in atmospheric moisture transport contributions associated with climate warming in the North Atlantic region Proposal for publication in Nature: Communications.
ROUND 1 (Jan/Feb 2023)

REVIEWER 2

We thank the reviewer for the work done on our manuscript. His/Her comments, questions, and suggestions have helped to improve it substantially.

The reviewer's main concern —regarding the short 5-year simulation periods used in the original manuscript— has been addressed. Despite the great (computational) effort required, the simulation periods have been extended to 30 years. Now, therefore, the results are much more robust, and the conclusions have greater reliability.

All other comments and suggestions made by the reviewer —which have also helped greatly in improving the article— are answered point by point below.

Fernández-Álvarez et al.

1- My largest concern on this work is that the results of this study are based on only 5-years of simulations for current and future climate. This is a very short period to draw conclusions from, taking into account the internal variability of our climate. By only analysing five years, it could happen that your results are biased to one or two dry or wet years being present in your data. I argue it is needed to extend the simulations of moisture sources before solid conclusions can be drawn. In terms of comparing climatologies of present and future climate, it's common to work with 30-year simulations, so the mean sources can be compared between two climates. By showing distributions of moisture sources, or performing significant tests between present and future climate, one can then address if the changes found in a future climate are actually significantly different. Now increases or decreases in sources can also happen from large anomalies in the 5-year data, instead of being an actual difference between present and future climate. While the limitation regarding using one climate model is mentioned under Limitations of this study, the shortness of the simulations being 5 year present and future climate is not mentioned in the manuscript at all, while I think it is a major issue regarding the methodology (and the related conclusions). I realise that it is a major effort to lengthen the simulations by 6-fold (for each scenario and period) to get robust results from 30-years of simulations, but it is needed to prove climate signal over internal variability.

We fully agree with the reviewer. The reason why we had initially provided only 5 years of simulations is none other than the computational limitations we had at that time. Fortunately, we have managed to expand our computational capacity, and the latest version of the manuscript is written on the basis of 30 years of simulation in all cases.

Specifically, the periods used in this new version of the manuscript are:

- Historical period: 1985-2014.
- Mid-Century (MC): 2036-2065.
- End-Century (EC): 2071-2100.

The original 5-year periods have been kept only to make a comparison between the model used and a BIAS-corrected ensemble; to answer a question asked by another reviewer.

2- The authors mention that they are the first to assess moisture sources using a Lagrangian tracking method in a future climate, on which I agree. However, similar studies have been performed with moisture tracking tools which use an Eulerian moisture tracking approach, and also applied this on future climate simulations (Findell et al. 2019 & Benedict, 2021). While the methodology to perform the tracking is different in those studies, the output product of the moisture source is similar, as is shown in earlier comparison studies of Eulerian and Lagrangian moisture tracking techniques (Van der Ent, 2013). The results from these earlier studies are in line with the results presented in this study as I explain below. Findell et al (2019) studied global moisture recycling in a future climate and concluded that oceanic moisture sources will become more important in the future. This is a similar conclusion as drawn in this study. The same conclusion, that moisture sources will become more important in a future climate was given by Benedict et al (2021), although Benedict et al only focused on the Mississippi basin. Specifically on the Iberian Peninsula, Findell et al. (2019) reports an increase in precipitation

recycling ratio towards the future, which is also concluded in the findings of this paper. Thus, although the exact technique to perform the tracking is not applied before on future simulations, the results drawn in this study are not necessarily surprising, but confirm findings from earlier studies as is also stated at multiple locations within the manuscript. Therefore I wonder if the results presented here show enough novelty to be published in this high-impact journal. Of course this work specifically focuses on the Iberian Peninsula, and the closely related surrounding atmospheric moisture sources. However, this is now not well expressed in the title of the manuscript. Therefore, I would suggest to include the region in the title of the manuscript, as the title now suggests a global study, which is not true. For example the title could be changed to: *Projected changes in moisture sources and sinks of the Iberian Peninsula associated with climate warming*.

We understand the doubts raised by the reviewer, and we will do our best to answer all of them.

Firstly, as the reviewer has pointed out, this study analyzes the future changes in the moisture source–sink relationship around the Iberian Peninsula, which is one of the hotspot mid-latitude areas affected by more than one moisture source (NATL and MED). Nevertheless, we also analyzed future changes of the moisture sinks associated with these moisture sources (MED and NATL) not only for the IP region, but for other regions of North Africa, Western and Eastern Europe, and Eastern North America.

It should be noted that the Mediterranean region itself is a hotspot for climate change (Lionello and Scarascia, 2018). Consequently, adaptation to changing climate threats is paramount for countries located around the Mediterranean Sea (Cramer et al., 2018), which live in a complex and diverse socioeconomic situation. On the other hand, the NATL source is the area that provides the highest moisture amount for precipitation over the surrounding continents (Gimeno et al., 2010). Demonstrating that it is one of the main physical mechanisms that transport moisture to Europe during winter (Eckhardt et al., 2004). Therefore, we consider that it can be said that our study presents an analysis of future changes in moisture sources and sinks for the North Atlantic region and its implications in areas of social interest.

That said, it is true that the original title was too pretentious, as our study is localized to a specific region. With this in mind, we ask the reviewer to accept the following proposal for modification of the title of the manuscript: *Projected changes in atmospheric moisture transport contributions associated with climate warming in the North Atlantic region*.

3. This study is analysing the changes in moisture sources towards the Iberian Peninsula, and the changes in precipitation from the source of the North Atlantic and Mediterranean. For me, the starting point of such analyses is to first look at the simulated precipitation itself. By doing so, the interpretation of the absolute moisture sources, increasing or decreasing, can be put in the perspective of decreasing or increasing precipitation in a changing climate. Right now, it remains unclear to the reader how the precipitation itself is changing towards the future, as only the moisture source results are discussed. I would suggest to start the result section by showing the changes in precipitation over the Iberian Peninsula (and the other land regions affected by the Mediterranean and North Atlantic) from the CESM model. Related to my previous comment on showing absolute precipitation changes, this will also benefit the interpretation on the precipitation contribution (PC) from oceanic moisture source results. Now it remains unclear if the precipitation contribution change because there is a relative increase/decrease of precipitation contribution because there is an actual larger relative contribution from the oceanic source, or because the total precipitation in the region is changing. Or another reason is that the dynamics are changing shifting the precipitation sinks (from the oceanic sources) to different regions. It would be very insightful to get a better interpretation of the precipitation contribution results.

Based on the reviewer's suggestion, the following discussion has been included in the **Results** section:

Future projections in precipitation and geopotential height as per the CESM2 model

The study of changes in the moisture source regions must be contextualized within the framework of the general changes in the precipitation regime of the regions analyzed. Thus, Figure Sup. Fig. 7 shows the seasonal and annual changes in the cumulative amount of precipitation (in mm/day) for both MC (2036-2065) and EC (2071-2100) with respect to the historical period (1985-2014) for the region of interest.

The most notable increases in precipitation are observed in mid- and high latitudes—as well as along the entire North American East Coast—for the winter months. Significant increases are also observed in the Gulf of Mexico region and particularly in Bermuda for the fall months. On the other hand, decreases in precipitation are characteristic of (sub)tropical latitudes for almost the entire year. In annual terms, a decrease in precipitation, close to 1 mm/day for EC, is observed over the entire Iberian Peninsula and the Mediterranean Sea. Decreases of more than 2 mm/day are observed in the southwestern sector of the study domain. On the other hand, increases close to 1 mm/day are expected in almost the entire east coast of the USA and Canada, as well as in the Bermuda region. In general terms, the patterns of precipitation changes are coincident between MC and EC; more accentuated in the latter.

Previous studies (Pereira et al., 2021; Vautard et al., 2014; Maule et al., 2017) already proposed a reduction in average precipitation for southern Europe. Specifically —and based on these previous studies— the expected precipitation reduction for the IP ranges from 10% to 15% for all seasons except winter. These changes would be in terms of a reduction in the number of wet days and an extension of drought periods by the end of the century (IPCC, 2021).

It should be taken into account that a determining factor in the changes of the precipitation regime -and the associated sources of humidity- may be the variations in the atmospheric dynamics of the future climate, with regard to the present one. In order to infer these, a study analogous to the one presented for accumulated precipitation in Figure Sup. Fig. 7 is presented for the geopotential height at 500 hPa (Z500) in Figure Sup. Fig. 8. In this regard, what is observed is a complex pattern, which can be summarized as a generalized increase of Z500 in the study region. The areas most affected by this increase are the subtropical regions, as well as the Atlantic coast of Canada and the northeastern United States. The areas least affected in annual terms —and particularly in the winter months— are the oceanic regions adjoining the British Isles. These results are in correspondence to those shown by Christidis et al. (2015) where using seven climate models demonstrated that a significant global increase in the annual and seasonal mean Z500 is projected. Finally, it is noteworthy that this generalized —albeit moderate— increase in mid-level pressures for this particular region has been identified in the literature as a potential trigger for the easing of the westerly flow over the North Atlantic, which could affect the position of the jet stream located over this region (Christidis et al., 2015).

Supplementary Figure 7. Comparison of average seasonal precipitation field between historical period and SSP5-8.5 scenario for CESM2 model. | Precipitation (in mm day⁻¹) for the historical reference period (left column, 1985–2014) and differences with the SSP5-8.5 scenario for the mid- and end 21st century (central, MC, 2036–2065, and right, EC, 2071–2100, respectively).

Supplementary Figure 8. Comparison of Geopotential height at 500 hPa field between historical period and SSP5-8.5 scenario for CESM2 model. | Same as Supplementary Figure 7 for geopotential height at 500 hPa (in km).

4- The term precipitation contribution is not a term which is familiar to the wider hydro meteorological community that can be reached with this article. If such a relatively unfamiliar term is introduced it requires a good explanation to be able to have a good interpretation of the results of this variable. Now, the results feel like an enumeration of findings on increases and decreases in the precipitation contribution, but it is not clear what the implications are of these findings.

In the introduction section, where the term is first cited, the text has been modified as follows:

(...) an increase in the recycling processes for the Iberian Peninsula, and a projected decrease in the precipitation contribution —term that will be discussed in the following sections— from the MED (...)

In the corresponding section, the term has been introduced as follows:

Changes in precipitation contribution from oceanic moisture sources

Beyond the analysis carried out for the Iberian Peninsula, the forward trajectories from the oceanic moisture sources (NATL and MED) were tracked to evaluate future changes in their precipitation contributions (PCs, contribution of a given source to the reanalyzed estimated total precipitation over a given sink (e.g., Nieto et al., 2019)) over the surrounding continental areas (see Methods section). Figure 3 (left column) provides regions with $P-E > 0$ having quantified the moisture balance for the cells originating in MED for the historical period. These PC patterns shows that MED provides a similar contribution over the continent adjacent to both the north and east of the MED basin in winter and fall (Fig. 3a, j). In the warm season, the most intense PCs move westward and have a greater effect during spring on Europe and during summer on Africa (Fig. 3d,g). These results agree with those of previous studies (Gimeno et al., 2010; Nieto et al., 2010).

5- The discussion as presented in the paper at the moment repeats many of the results presented before, and here and there, connects those results with the existing literature. To me, it feels as an almost complete repetition of the results with some extra interpretation. This makes the discussion very lengthy and not to the point. I would suggest to either shorten the summary of the results in this discussion section and emphasize the real discussion part. Or to move the discussion of the results directly to the result section and only provide a short summary and outlook in the last section which you could call synthesis. In general, I would like to see more of an outlook of the implications of your results as now in the discussion mostly numbers and connection with existing literature is given, while I would like to hear more about what these results in the field of precipitation, land-atmosphere

interactions etc. This is what happens a little bit at the end of the discussion section, but is not highlighted very clearly because of the lengthy section.

In order to comply with the reviewer's request, the results section has been completely rearranged.

The section "subsections" are now arranged as follows:

1. Future projections in precipitation and geopotential height as per the CESM2 model
2. Future projections for integrated water vapour transport (IVT) in the North Atlantic Ocean
3. Changes in moisture sources for the Iberian Peninsula
4. Changes in precipitation contribution from oceanic moisture sources
5. Summary

6- Line 47-48: Changes in moisture recycling in the Iberian Peninsula. Are the numbers you give here based on the research of this paper? If so, I would mention it only later in the conclusion. If not, it needs a reference, and then you can compare it to the numbers you get in this study.

Certainly, the results cited by the reviewer correspond to Findell et al. (2009). The reference has been included in the text, and these results have also been compared with ours in the summary section of the latest version of the manuscript.

Summary

The hydrological cycle is projected to increase in intensity under climate change (Sun et al., 2007). We found a general increase in moisture reaching continental areas in the extratropical North Atlantic and Mediterranean belts. We also observed a significant increase—although to a lesser extent— of PRPs over the Iberian Peninsula in the winter months. In annual terms, we observed an increase in these PRPs of between 2 and 8%, within the range previously established by Findell et al. although these are doubled for EC. In any case, the projections presented in this article coincide with those that had already identified an increase in the contribution of moisture from oceanic sources—and, to a lesser extent, from the recycling processes—with global warming (Langenbrunner, 2019).

(...)

7- Line 57-58: Can you supply some more motivation why you focus on the Iberian Peninsula? And why the MED and NATL are defined as they are? (for the MED region it is obvious, but the reason for selecting such a specific NATL region is not clear to me)

The Iberian Peninsula was selected as the region featured in the study with the following motivation:

1. The IP is one of the hotspot mid-latitude areas affected by more than one moisture source (Gimento et al., 2012).
2. This region is optimal for conducting this kind of analysis since its climate is highly dependant on the intrusion of moisture from NATL and MED (Gimeno et al., 2010b; Nieto et al., 2010) as well as affected by heavy recycling processes.
3. The IP is one of the areas with the highest frequency of Atmospheric Rivers in the North Atlantic along with the British Isles (e.g., Algarra et al., 2020).

This motivation has been included in the text of the latest version of the manuscript as follows:

Specifically, we analysed future changes in the moisture source–sink relationship around the Iberian Peninsula (IP), which is one of the hotspot mid-latitude areas affected by more than one moisture source (Gimeno et al., 2012) (Supplementary Fig. 1). This region is optimal for conducting a detailed study since its climate is highly dependent on the moisture intrusion from two of the major global oceanic sources (Gimeno et al., 2010; Nieto et al., 2010) (Supplementary Fig. 1), the North Atlantic Ocean (NATL, main source of moisture for neighboring continents in winter months (Eckhardt et al., 2004; Gimeno et al., 2010)) and the Mediterranean Sea (MED, known to be particularly sensitive to climate change and its implications (Lionello and Scarascia, 2018; Cramet et al., 2018)). It is also affected by strong recycling processes (Gimeno et al., 2010) (see Supplementary Section 1.3 for more information) and—along with the British Isles—the IP is the most active region in the North Atlantic for atmospheric river activity (Algarra et al., 2020).

The scope of this study—although giving greater relevance to the Iberian Peninsula— reaches all the regions that have NATL and MED as a relevant source of moisture; North Africa, Western and Eastern Europe and Eastern North

America. The main means to perform this analysis is the use of a Lagrangian dispersion model initialized with historical data and with a global model incorporated in the CMIP for the SSP5-8.5 scenario (O'Neill et al., 2016; Riahi et al., 2017). (see Data Description in Supplementary Section 1.1) (...)

As for the selection of such an specific NATL region; the reason for this selection is consistency with previous work conducted by our group (and others). Particularly, with the methodology usually applied by us, moisture source regions are bounded by the maximum values in integrated moisture flux divergence (Gimeno et al., 2010b). These fields are calculated from ERA5 throughout 1980-2018. The mask used is not arbitrary, but has been chosen formally; particularly with the 750 mm/yr threshold at divergence from oceanic sources. The masks used in this research have been previously used in the literature on several occasions. Particularly, NATL has been introduced by Gimeno et al. (2010b); and MED has been introduced by Nieto et al. (2010).

8-Lines 65-66: You should mention in the main text what is the time period that you have done the analysis for and how that influences your results (see major point on methodology)

With the extension of the simulations to 30 years, and the use, therefore, of a climatic period, this issue has been solved.

9- Line 65: global climate models ->should be global climate model as you only use one model, have you checked if this model gives a good representation of precipitation and circulation around the Iberian Peninsula? That could be a verification for using the CESM model. Now it is unclear why you have picked that specific model to do the analyses for.

The selection of the CESM2 to carry out this research was based mainly on two criteria:

- i. **The CESM2 has been previously evaluated for the variables involved in moisture transport.** Specifically, CESM2 data has been evaluated for representing jet streams and storm tracks, Northern Hemisphere (NH) stationary waves, global divergent circulation, annular modes, the North Atlantic Oscillation and NH winter blocking (Simpson et al., 2020). CESM2 ranks within the top 10% of CMIP class models with respect to many of these features (Simpson et al., 2020). In addition, precipitation prediction in CESM2-based subseasonal forecast systems has been shown to be similar to the prediction of the NOAA CFSv2 model, and slightly lower than the prediction of the ECMWF model (Richter et al., 2022). Besides, the NAO —critical to our region of interest— structure in winter and summer is relatively well represented in CESM2 with some minor biases that are quite similar to the rest of the CMIP6 climate models (Simpson et al., 2020). This implies an adequate representation of the associated precipitation anomalies over the Mediterranean (Blade et al., 2012).

On the other hand, CESM2 has a remarkable representation of the velocity potential of the upper troposphere in both summer and winter. This element is closely related to tropical precipitation and represents a significant forcing of extratropical standing waves (Simpson et al., 2020). Moreover, it presents improvements in rainfall in regions of great global interest such as the Indian Ocean, East Asia, the tropical Atlantic and the Amazon (Simpson et al., 2020). Finally, CESM2 has been used to study the sea surface temperature effect increase on future changes in Atmospheric Rivers, which is an important mechanism in moisture transport from tropical regions to mid and high latitudes (Gimeno et al., 2016, McClenny et al., 2020, 2021).

- ii. **CESM2 is a very convenient model for initializing WRF simulations.** In this regard, it should be emphasized that not only are all the variables needed to initiate WRF-ARW provided directly by CESM2, but also the natural resolution at which these variables are provided is very good. While other models provide natural resolutions of 250 km, CESM2 works with a resolution of 100 km, which has been very useful in previous related work such as, for example, for the simulation of precipitation over the US West Coast (Rahimi et al, 2021, 2022). If we had had to perform interpolations or even had to obtain certain necessary variables from others it would have added uncertainty to the results and would have multiplied the computational effort.

We let the reviewer know that as part of the response to Reviewer 1, we have compared 5 years of CESM2 simulations with those of a CMIP6 ensemble of 18 models. In that comparison we have shown that the response of CESM2 alone was reasonably good, and justified its use in this work. (More details in the response to Reviewer 1's question #2).

In any case, to answer question **have you checked if this model gives a good representation of precipitation and circulation around the Iberian Peninsula?** we provide here Figures Figure Rev2.3 and Rev2.4 with the comparison between the historical outputs of ERA5 and CESM2 for the mean precipitation and geopotential height fields at 500 hPa.

The reviewer can verify that in general —and particularly near the Iberian Peninsula— the variations in Z500 are not appreciable and the differences in the precipitation fields are less than 1 mm/day; especially in annual terms.

We consider, therefore, that the good performance of CESM2 is sufficiently proven and that in spite of its limitations (discussed in the manuscript) it fits well within the purpose of this work.

Figure Rev2.3

Figure Rev2.4

10- Line 99-101: I think it is also good to link to research about the storm track here, as that is basically what you are analysing, moisture transport within extra-tropical cyclones.

The following paragraph has been included in the “summary” section:

This displacement will be clearly related to the shift in the trajectories of extratropical cyclones, since these are the main mechanism of this poleward transport of moisture (Villamil-Otero et al., 2018). This potential shift, which has already been observed in recent decades, is also expected to intensify under less optimistic climate change scenarios (Bender et al., 2012).

11- Line 202-203: Here you mention a ‘decrease in recycling ratios’ while the conclusion was just that there was an increase in recycling ratio over the Iberian Peninsula (line 189-190). This seems very contradictory me, can you please comment and also verify in the manuscript.

Certainly, the sentence contained in lines 202-203 of the original manuscript was not correctly worded. That part of the text has been completely rewritten according to the new extended simulation period.

12- Supplementary Figures 10 & 11, and other similar figures: Can you combine the information in those figures by giving percentages of source contributions per region in barplots, given the different scenario’s and timesteps. In this way it will be much easier to compare the different simulations.

By extending the simulations to 30 years, and focusing the study on the SSP-8.5 scenario, the context in which the reviewer has made this suggestion has completely changed.

We kindly ask the reviewer to check the new figures in the supplementary material within their new context, and let us know if he/she continues to detect any problem.

13- It is very hard to give good comments on the figures if the caption are not directly with the figure (now you have to count the number of figures and then relate it to the caption). Besides, I think some figures in the main manuscript are actually figure which belong to the supplementary figures (as the supplementary material does not contain any figure), which is very confusing and makes it hard to provide good feedback on the figures.

We understand the reviewer’s discomfort in this regard. It is not always easy to adapt to journal templates and review systems.

The reviewer can check that in the new version of the manuscript the figures are attached next to the main text. A large number of the figures originally contained in the supplementary material have been removed when considering the SSP-8.5 scenario only.

14- The splitting up in seasons regarding the figures with the barplots (which I think do present the results in a concise manner) is to me a bit illogic as I use to think in seasons: DJF, MAM, JJA, SON. Is there a reason for the authors to have the seasons like this, and could you support this?

Certainly, we are all used to the seasons used in the analysis being DJF, MAM, JJA and SON.

However, in this case we have chosen JFM, AMJ, JAS and OND for two reasons:

1. Firstly, this has allowed us to make good use of the 30 years without any temporal fractures in the continuity of the simulation. These simulations are very expensive in computational terms, and we felt that this was the best way to take advantage of them.
2. Second, and more importantly, this study is part of others that we are conducting within the context of a larger project. Other papers already published by our group (e.g., Gimeno et al., 2010a; Fernandez-Alvarez et al., 2022) use these same seasons and we prefer to maintain consistency so that the results can be compared.

We ask the reviewer to understand these reasons, and to allow us to keep the seasons as originally planned.

References

- Algarra, I., Nieto, R., Ramos, A. M., Eiras-Barca, J., Trigo, R. M., & Gimeno, L. (2020). Significant increase of global anomalous moisture uptake feeding landfalling atmospheric rivers. *Nature communications*, 11 (1), 1–7.
- Benedict, I., Van Heerwaarden, C. C., Van Der Ent, R. J., Weerts, A. H., Hazaleger, W. Decline in Terrestrial Moisture Sources of the Mississippi River Basin in a Future Climate. *J. Hydrometeorol.* 21, 299-316 (2020).
- Bender, F. A.-M., V. Ramanathan, and G. Tselioudis, 2012: Changes in extratropical storm track cloudiness 1983–2008: Observational support for a poleward shift. *Climate Dyn.*, 38, 2037–2053, <https://doi.org/10.1007/s00382-011-1065-6>.
- Cramer, W., Guiot, J., Fader, M., Garrabou, J., Gattuso, J. P., Iglesias, A., Lange, M. A., Lionello, P., Llasat, M. C., Paz, S., Peñuelas, J., Snoussi, M., Toreti, A., Tsimplis, M. N., and Xoplaki, E.: Climate change and interconnected risks to sustainable development in the Mediterranean. *Nat. Clim. Change* 8, 972–980 (2018).
- Chiang, F., Mazdiyasi, O., & AghaKouchak, A. (2018). Amplified warming of droughts in southern United States in observations and model simulations. *Science advances*, 4(8), eaat2380.
- Christidis, N., Stott, P. A. Changes in the geopotential height at 500 hPa under the influence of external climatic forcings. *Geophys. Res. Lett.* 42, 10-798 (2015).
- Findell, K. L., Keys, P. W., Van Der Ent, R. J., Lintner B. R., Berg, A. & Krasting, J. P. Rising Temperatures Increase Importance of Oceanic Evaporation as a Source for Continental Precipitation. *J. Climate* 32, 7713-7726 (2019).
- Eckhardt, S., Stohl, A., Wernli, H., James, P., Forster, C., Spichtinger, N. A 15 year climatology of warm conveyor belts. *J. Clim.* 17, 218–237 (2004).
- Fernández-Alvarez, J. C., Vázquez M., Pérez-Alarcón A., Nieto R., Gimeno L. (2022) Comparison of moisture sources and sinks estimated with different versions of FLEXPART and FLEXPART-WRF models forced with ECMWF reanalysis data. <https://doi.org/10.1175/JHM-D-22-0018.1>
- Gimeno, L., Nieto, R., Trigo, R., Vicente-Serrano, S.M. & López-Moreno, J. I. Where does the Iberian Peninsula moisture come from? An answer based on a Lagrangian approach. *J. Hydrometeorol.* 11, 421-336 (2010a).
- Gimeno, L., Drumond, A., Nieto, R., Trigo, R. M. & Stohl, A. On the origin of continental precipitation. *Geophys. Res. Lett.* 37, L13804 (2010b).
- Gimeno, L., Stohl, A., Trigo, R. M., Dominguez, F., Yoshimura, K., Yu, L., Drumond A., Durán-Quesada, A. M. & Nieto, R. Oceanic and terrestrial sources of continental precipitation. *Rev. Geophys.* 50, RG4003 (2012).
- King, M., Altdorff, D., Li, P., Galagedara, L., Holden, J., & Unc, A. (2018). Northward shift of the agricultural climate zone under 21st-century global climate change. *Scientific Reports*, 8(1), 1-10.
- Lionello, P., Scarascia, L. The relation between climate change in the Mediterranean region and global warming. *Reg. Environ. Change.* 18, 1481–1493 (2018).
- Lehtonen, I., Ruosteenoja, K., & Jylhä, K. (2014). Projected changes in European extreme precipitation indices on the basis of global and regional climate model ensembles. *International journal of climatology*, 34(4), 1208-1222.
- Langenbrunner, B. Shifting moisture source. *Nat. Clim. Chang.* 9, 728 (2019). Nieto, R., Gimeno, L., Drumond, A. & Hernandez, E. A Lagrangian identification of the main moisture sources and sinks affecting the Mediterranean area. *WSEAS Trans. Environ. Dev.* 6, 365-374 (2010).
- Nieto, R., Ciric, D., Vázquez, M., Liberato, M. L., & Gimeno, L. (2019). Contribution of the main moisture sources to precipitation during extreme peak precipitation months. *Advances in Water Resources*, 131, 103385
- Villamil-Otero, G. A., Zhang, J., He, J., & Zhang, X. (2018). Role of extratropical cyclones in the recently observed increase in poleward moisture transport into the Arctic Ocean. *Advances in Atmospheric Sciences*, 35(1), 85-94.
- Xu, Z., Han, Y., Tam, C. Y., Yang, Z. L., Fu, C. Bias-corrected CMIP6 global dataset for dynamical downscaling of the historical and future climate (1979–2100). *Scientific Data* 8, 1-11 (2021).
- Yin, J. H., 2005: A consistent poleward shift of the storm tracks in simulations of 21st century climate. *Geophys. Res. Lett.*, 32, L18701, <http://dx.doi.org/10.1029/2005GL023684>.

REVIEWER COMMENTS

Reviewer #1 (Remarks to the Author):

Projected changes in atmospheric moisture transport contributions associated with climate warming in the North Atlantic region

Manuscript ID: NCOMMS-22-17469

I am pleased to see that authors have addressed my previous comments elaborately and reasonably. I am aware that the additional work was computationally heavy, but it certainly improved the study scientifically. I congratulate the authors on their efforts, and I recommend publication of the manuscript.

I noticed a few typos in the text that could use some editing. I pointed them out below;

Typo

1- In Introduction section -> neighboring -> neighbouring

2- Section "Future projections for integrated water vapour transport (IVT) in the North Atlantic Ocean" -> analized -> analysed

3- Title "Changes in precipitation contribution from mediterranean and north Atlantic moisture sources" -> the Mediterranean

4- Section "Changes in precipitation contribution from mediterranean and north Atlantic moisture sources" pg. 14 -> behaviors -> behaviours

5- pg. 16 -> atlantic -> Atlantic

6- pg. 16 -> southeastern -> south-eastern

5- In Summary section -> This changes -> These changes

Reviewer #2 (Remarks to the Author):

Review round 2 on Projected changes in atmospheric moisture transport contributions associated with climate warming in the North Atlantic region

First, I like to thank the authors for addressing my comments. The new title is more suitable and tailored to the work that has been done. I am also happy to see that the simulation time has been extended from 5- to 30-years per 'climate', which is an impressive amount of work done, and which was needed to make fair conclusions. This has solved my major concern on simulation length, and comparing climates. In addition, a more clear outline was given on why the authors have chosen to perform the tracking with the CESM model (point raised by another reviewer) and this has also been rectified based on comparison of the historical period with ERA5. I understand that the performed analyses cannot be extended to multiple climate models, although that would be something to aim for in the future. The authors also addressed the general changes in precipitation and circulation based on the CESM model, which provides more context at the start of the paper, to be able to interpret the results on the moisture sources afterwards. While the results on precipitation and circulation are discussed in depth, I miss some further interpretation on why and how the moisture sources and sinks are changing the way they do. My main comments are related to the interpretation and description of the moisture source and precipitation contribution results:

1) For me it is still a bit unclear what PCs exactly is, as it is described to be a percentage (a contribution of something to something; line 196-197), where the values in the figure show absolute numbers of E-P (so I think there is no contribution to the total precipitation taken, or am I mistaken?). See also my comments below regarding line 196-197.

2) Related to my point on precipitation contributions I am wondering if the precipitation contributions in absolute sense increases to a future climate for the MED and ATL region, as evaporation from these oceanic regions is likely increasing. If you track more moisture from the start

you would also expect more sinks, or is there no more moisture tracked? It would be nice if this can be reflected upon: How is the SST/evaporation changing towards the future in the CESM downscaled model, which you are then tracking forward in time if I understand correctly, so that will lead to increased precipitation contributions overall in the future? Also would this mean that if you see a decrease in PC, for example from the Mediterranean source in AMJ over middle of Europe, that this is related to dynamic changes, as if you would purely look to same dynamics and more moisture in the air, you would expect an increase? This question/comment also related to my next point.

3) What I am still missing is further interpretation of the results on why sources and sinks are changing. The interpretation on changes in precip and circulation (500 hPa geopotential height) are put nicely in perspective by referring to literature and discussing. However, for the moisture sources this discussion and interpretation is not given in the results, but is discussed in the summary, although very limited. I think the section of the summary is not the correct name for that section, it should either be Synthesis, or the discussion should be moved to the results section directly and your summary will be much shorter (that would have my preference). Point 2 is also related to the discussion of the results.

4) I know there is limited space for text but the methodology in the article itself is now very limited and I would like to see some additional information in the paper explaining what EC and MC is, for which years it hold, which GCM you used and that you downscaled this GCM with WRF, and which moisture tracking models you used. Details on model settings and so on can be given in the appendix, but the names of the models and approaches and time span should be mentioned in the main article in my opinion.

Smaller comments (please note that I have given my comments based on the file with track changes as that file contains line numbers)

Line 36-37: In addition to warming, an increase in insert(global) mean annual evaporation is expected, remove(which will affect) insert(affecting) the evaporation rates of most oceans.

Line 48-49: With the word this (This projected decrease) it seems like you refer to the previous sentence where you are talking about an increase, which is a bit confusing to read.

Line 70-71: The main means to perform this analysis > The analysis are performed using a Lagrangian

..

Line 68-76: As indicated as main comment, I would prefer more detailed information here regarding which models you used, mention that you performed a downscaling, which timeframes the analysis were done for etc.

Line 88: MC and EC have not been discussed before so write them out for the first time, also I would prefer that the timeframes were introduced in the methods section in the introduction.

Line 89: region > regions

Line 138; Therefore is an illogic connecting word here for me

Line 153: over the remove(proper) region

Line 147-154: could you give some numbers here to make it more quantitative?

Figure 1: I would put annual on top in all those figures, and discuss the annual results before moving to the different seasons.

After line 186: After reading this results I am very much wondering about the interpretation of these results. Why is the local contribution decreasing in MC and increasing in EC? Do you have any ideas? Putting perspective here would add insights into this paper.

Line 195: were remove(tracked) insert(used) to evaluate

Line 196-197: you use the word reanalysed estimated total precipitation which makes me think of reanalysis products like ERA5 but that's not what you mean here I believe. So what do you mean with reanalysed? Also given the explanation given for PC I would expect a percentage as its mentioned 'a contribution of something to something', but in fact the figure shows absolute numbers which I found confusing. Summarized, I am still not sure what PC exactly entails and I think it is important to make this clear as this paper will be interesting to read for a broad audience who might not all be familiar with the concept of moisture tracking and PCs.

Line 206: show results that depend on the season remove(of the year) considered

Line 208: and insert(this is) particularly strong

Line 214: MCs > ECs

Line 266: ...for Figure 3, but now for the NATL

Figure 5: In the right column I see this shift in negative and positive contributions over the North Atlantic, especially in winter (JFM, OND), and I was wondering why this happens and if you can discuss (now it is not explicitly mentioned in the text I believe). Is this related to the shift in the location of the storms tracks that you mention earlier?

Line 298-299: main responsible > largest contributor to this increase

Line 300-302: Here you discuss the storm tracks, nice, is this the explanation of my question for Figure 5?

Figure 2, 4 & 6: Is it possible to add an estimate of variability to this plot? As you now have 30 years of tracking per period, the yearly and seasonal variability can be given as a standard deviation for example. An indication of variability would put the MC versus EC into perspective and also gives an indication of the small positive absolute numbers. For example in Figure 6c a small number in EUwest annual, is this a small positive number in all 30 years, or is it sometimes also a negative relative contribution?

Line 307: To me the Summary is now more of a discussion + conclusion, thus if you keep the discussion into your summary I would choose a different title of this section

An interesting question that popped up in my head when reading the methods is if your travel time of moisture will change into a warmer climate, where the air holds more moisture. And if so, if you need to change your travel time accordingly yes or no. This is not a comment but more of a follow-up question/thought.

Line 424: The term WRF-ARW is new for me here, and I don't see how it relates to your work (the - ARW part, what does the abbreviation mean and what is the context?)

Projected changes in atmospheric moisture transport contributions associated with climate warming in the North Atlantic region.

Proposal for publication in Nature: Communications.

ROUND 2 (Apr/May/Jun 2023)

5

REVIEWER 1

I am pleased to see that authors have addressed my previous comments elaborately and reasonably. I am aware that the additional work was computationally heavy, but it certainly improved the study scientifically. I congratulate the authors on their efforts, and I recommend publication of the manuscript.

10 We are very grateful to the reviewer for him/her help in improving the manuscript. We certainly agree that the extension of the analyzed period was necessary.

We also appreciate the recommendation for publication of our manuscript.

Fernández-Álvarez et al.

1. Minor Comments

15 **I noticed a few typos in the text that could use some editing. I pointed them out below;**

Typo

1- In Introduction section ->neighboring ->neighbouring

2- Section "Future projections for integrated water vapour transport (IVT) in the North Atlantic Ocean" ->analized ->analysed

20 **3- Title "Changes in precipitation contribution from mediterranean and north Atlantic moisture sources" ->the Mediterranean**

4- Section "Changes in precipitation contribution from mediterranean and north Atlantic moisture sources" pg. 14 ->behaviors ->behaviours

5- pg. 16 ->atlantic ->Atlantic

25 **6- pg. 16 ->southeastern ->south-eastern**

5- In Summary section ->This changes ->These changes

All these typographical errors have been corrected in the latest version of the manuscript, following the recommendations of the reviewer. Once again, we are grateful for the help received in the preparation of this paper.

Projected changes in atmospheric moisture transport contributions associated with climate warming in the North Atlantic region
Proposal for publication in Nature: Communications.
ROUND 2 (Apr/May/Jun 2023)

REVIEWER 2

First, I like to thank the authors for addressing my comments. The new title is more suitable and tailored to the work that has been done. I am also happy to see that the simulation time has been extended from 5- to 30-years per 'climate', which is an impressive amount of work done, and which was needed to make fair conclusions. This has solved my major concern on simulation length, and comparing climates. In addition, a more clear outline was given on why the authors have chosen to perform the tracking with the CESM model (point raised by another reviewer) and this has also been rectified based on comparison of the historical period with ERA5. I understand that the performed analyses cannot be extended to multiple climate models, although that would be something to aim for in the future. The authors also addressed the general changes in precipitation and circulation based on the CESM model, which provides more context at the start of the paper, to be able to interpret the results on the moisture sources afterwards. While the results on precipitation and circulation are discussed in depth, I miss some further interpretation on why and how the moisture sources and sinks are changing the way they do. My main comments are related to the interpretation and description of the moisture source and precipitation contribution results.

We thank the reviewer for all his/her work, both throughout the entire process and in this second revision of the manuscript. We take this second opportunity to continue to improve the manuscript and to address all the reviewer's suggestions.

Attached below are the answers to the questions and suggestions that have been raised by the reviewer in this second round of review.

Fernández-Álvarez et al.

1- For me it is still a bit unclear what PCs exactly is, as it is described to be a percentage (a contribution of something to something; line 196-197), where the values in the figure show absolute numbers of E-P (so I think there is no contribution to the total precipitation taken, or am I mistaken?). See also my comments below regarding line 196-197.

The reviewer is not mistaken, this misleading is probably due to a lack of explicitness on our part. PCs are nothing more than the contribution in amount of total precipitation (presented as daily averages, i.e., mm/day) from the sources to the sink region. It is neither a recycling ratio nor a percentage value, but simply the rainfall contribution of the sources to the region of interest.

The section where this concept is introduced has been edited as follows, to avoid possible confusion in the reading of the manuscript:

Beyond the analysis carried out for the Iberian Peninsula, the forward trajectories from the oceanic moisture sources (NATL and MED) were used to evaluate future changes in their precipitation contributions (PCs, total mean amount of precipitation—in mm/day— provided by the sources to the study region³³) over the surrounding continental areas (see Methods section). Figure 3 (left column) provides regions with $P-E > 0$ having quantified the moisture balance for the air parcels originating from MED for the historical period. Overall, a remarkable PC is observed in northern and eastern Europe, as well as in regions of North Africa, with contributions of between 3 and 7 mm/day (Fig. 3a). In addition, these PC patterns show that MED provides a similar contribution over the continent adjacent to both the north and east of the MED basin in winter and autumn (Fig. 3d, m). In the warm season, the most intense PCs move westward and have a greater effect during spring on Europe and during summer on Africa (Fig. 3g, j). These results agree with those of previous studies^{17,18}.

2- Related to my point on precipitation contributions I am wondering if the precipitation contributions in absolute sense increases to a future climate for the MED and ATL region, as evaporation from these oceanic regions is likely increasing. If you track more moisture from the start you would also expect more sinks, or is there no more moisture tracked? It would be nice if this can be reflected upon: How is the SST/evaporation changing towards the future in the CESM downscaled model, which you are then tracking forward in time if I understand correctly, so that will lead to increased precipitation contributions overall in the future? Also would this mean that if you see a decrease in PC, for example from the Mediterranean source in AMJ over middle of Europe, that this is related to dynamic changes, as if you would purely look to same dynamics and more moisture in the air, you would expect an increase? This question/comment also related to my next point.

The question raised by the reviewer is challenging, and we will try to give the best possible answer. As we already know, an increase in evaporation (E) in NATL and MED sources is to be expected in the future, which will also be combined with an increase in SST. This will undoubtedly lead to a more intense transport of moisture that will have a direct impact on sinks. The way in which this impact occurs is determined by several dynamic factors that affect the necessary convergence of this moisture flux.

Specifically, a decrease in the contribution of the Mediterranean source has been observed for central Europe in AMJ, together with an increase in evaporation (Fig. Rev1) and SST (Fig. Rev2). We understand that this behaviour is associated with changes in the dynamics for the future. Likewise, in Fig. Rev3 there are noticeable changes in the moisture fluxes. These changes over the MED source region are most noticeable to the east with a predominantly southward flow, weakening further moisture transport towards Eastern Europe. A different behaviour is observed over North Africa (see Figure 4 in the manuscript) with both a seasonal and annual increase in the MED contribution. In addition, it was shown that there is an increase in the geopotential height values at 500 hPa for the CESM2 model over this region (similar result is expected for WRF forced with CESM2). This result shows regions of greater stability in the vertical column preventing greater contribution from this source to the total precipitation in this region. The largest positive differences are always observed at the EC (see Figure S8).

Therefore, we believe that it can be concluded that this region is mainly influenced by dynamical changes expected for the future climate that decrease the contribution from the MED and not as much from thermodynamic changes that lead to higher availability of moisture in the atmosphere.

The following paragraph has been included in the *Synthesis* section to discuss this idea:

The results obtained in relation to the increase in the contribution of NATL and MED moisture to the IP confirm that with the rise in temperature at the end of the century, the atmosphere will increase its water retention capacity in quantities in line with those predicted by the Clausius-Clapeyron equation⁴⁶ with $7\%K^{-1}$. This would imply an increased evaporation in the source regions such as the MED and NATL, with an accentuated moisture supply that, at least in part, would end up affecting the IP. This can be seen in Fig. Sup. 14, where for both MC and EC a very noticeable increase in evaporation is projected over NATL and, to a lesser extent, over MED. Moreover, they show a slight latitudinal shift from this source region to the EC. Finally, the results show a clear decrease in the contribution of the NATL region located above latitude 40N and close to the European coast, which may be related both to a considerable decrease in evaporation for MC and EC in this region (Fig. Sup. 14) and to the increase of the Z500 field which implies a higher anticyclonic circulation and therefore less atmospheric instability (Fig. Sup.8).

Higher contributions of the MED moisture source to precipitation are observed in the regions located south of the Mediterranean Sea, both for MC and EC. This contribution will also increase —although to a lesser extent— over the Iberian Peninsula, mainly in the summer season, being more noticeable for EC. There is also observed a decrease in the contribution in Eastern Europe and Western Europe (in summer and spring)^{47–49}. This result may be associated with the projected latitudinal shift of storm trajectories in the future climate⁴⁰. In addition, much of this behavior could be attributed to changes in atmospheric dynamics for both MC and EC. Fig. Sup. 13 shows that for both periods changes in the moisture flux patterns are to be expected. Over the MED region, changes are more noticeable in the eastern region where a weakening of moisture transport towards Eastern Europe is observed. On the other hand, reinforcements of Z500 are observed suggesting that in the future, the anticyclonic circulation and consequently the atmospheric stability as well as the blocking of the fluxes from the sources in these regions will be greater. This evidence indicates that these changes in atmospheric dynamics may be as or more relevant than the thermodynamic changes that lead to greater availability of moisture in the atmosphere. Further, these results show correspondence with those obtained by Batibenz et al.⁵⁰ over Eastern Europe, showing negative trends for seasonal precipitation trends per year using four reanalyses in summer.

Figure Rev1. Comparison of SST field between historical period and SSP5-8.5 scenario focused on the MED | SST field (coloured filled, in °C) for the historical period (left column, 1985–2014) and associated differences for the SSP5-8.5 scenario for MC and EC in the middle and right column, respectively. FOR THE REVIEW PROCESS, NOT INCLUDED IN THE MANUSCRIPT.

Figure Rev2. Same as Figure Rev1, but for evaporation in mm/day.
 FOR THE REVIEW PROCESS, NOT INCLUDED IN THE MANUSCRIPT.

Figure Rev3. Same as Figure Rev1, but for IVT in $\text{kg m}^{-1} \text{s}^{-1}$
 FOR THE REVIEW PROCESS, NOT INCLUDED IN THE MANUSCRIPT.

Figure Rev4. Same as Figure Rev2, but including the entire study region.
 FOR THE REVIEW PROCESS, NOT INCLUDED IN THE MANUSCRIPT.

3- What I am still missing is further interpretation of the results on why sources and sinks are changing. The interpretation on changes in precip and circulation (500 hPa geopotential height) are put nicely in perspective by referring to literature and discussing. However, for the moisture sources this discussion and interpretation is not given in the results, but is discussed in the summary, although very limited. I think the section of the summary is not the correct name for that section, it should either be *Synthesis*, or the discussion should be moved to the results section directly and your summary will be much shorter (that would have my preference). Point 2 is also related to the discussion of the results.

The cited section has been renamed as *Synthesis*. For the rest of the discussion, please refer to the previous question.

4- I know there is limited space for text but the methodology in the article itself is now very limited and I would like to see some additional information in the paper explaining what EC and MC is, for which years it hold, which GCM you used and that you downscaled this GCM with WRF, and which moisture tracking models you used. Details on model settings and so on can be given in the appendix, but the names of the models and approaches and time span should be mentioned in the main article in my opinion.

In response to the reviewer's request (which we consider reasonable), the manuscript has been modified to include more detailed information on the methodology developed and the tools used. In particular:

Section *Data Employed*:

The available outputs of the climate model Community Earth System Model Version 2 (CESM2)⁵⁹, from Phase 6 of the Coupled Model Intercomparison Project (CMIP6), were dynamically downscaled. The CESM2 data were downloaded from the Earth System Grid Federation (ESGF2) and obtained for the native "gn" grid with a resolution of 0.9 x 1.25 (≈ 1°) presented as an output mesh with 288 × 192 longitude/latitude, 32 vertical levels (top level at 2.25 mb). Specifically, Weather Research and Forecasting model (WRF-ARW) in its version 3.8.1⁶⁰ was used to downscale CESM2 data (see Supplementary Section 1.2). The highest shared socioeconomic pathway (SSP) scenario from the CMIP6 climate projections (SSP5-8.5) has been also

used to analyse the different ranges of future forcing pathways to 2100 (see Supplementary Section 1.1). A set of 30-year periods were compared spanning the historical period 1985–2014 and the intervals 2036–2065 (for the mid-century: MC) and 2071–2100 (for the end-century: EC). ERA5⁶¹ reanalysis data were also used to evaluate the results for the historical period. ERA5 is chosen as a reference since this model provides the advantages of high resolution (31 km horizontally and 137 levels vertically) and a large number of assimilated historical observations. In addition, ERA5 significantly improves upon its predecessor, ERA-Interim reanalysis, particularly with respect to precipitation fields both over extratropical regions and tropical oceanic areas. A more detailed description is presented in the Supplementary Information.

WRF-ARW and FLEXPART-WRF setups

The parameterisations employed in the WRF-ARW configuration were as follows: the WSM6 microphysics scheme⁶², Yonsei University planetary boundary layer (PBL) scheme⁶³, revised MM5 surface layer scheme⁶⁴, United Noah Land Surface Model⁶⁵, shortwave and longwave RRTMG schemes⁶⁶ and the Kain-Fritsch Ensemble cluster scheme⁶⁷. The WRF-ARW outputs had 40 vertical layers from the surface to 50 hPa with a horizontal spacing of 20 km and they covered an area of 115.39–42.02W and 19.41S–59.51N (see Fig. Sup. 1). For the FLEXPART-WRF⁶⁸ configuration, we used Hanna's⁶⁹ scheme for turbulence parameterisation with the convection scheme activated. This scheme is based on the boundary layer parameters PBL height, Monin–Obukhov length, convective velocity scale, roughness length and friction velocity⁶⁸. We assumed skewed rather than Gaussian turbulence in the convective PBL. The FLEXPART-WRF has forty levels and 400 x 777 points, where in the output mesh where the particles are released. The outputs had spatial and temporal resolutions of 20 km and 6 h, respectively.

MINOR COMMENTS

5- Line 68-76: As indicated as main comment, I would prefer more detailed information here regarding which models you used, mention that you performed a downscaling, which timeframes the analysis were done for etc.

The following descriptions have been included in the Introduction:

(...) Eastern Europe and Eastern North America. The analysis is performed using the Lagrangian dispersion model FLEXPART-WRF initialized with WRF-ARW dynamically downscaled outputs. Three representative climatic periods have been used for the comparison. Firstly, a historical period (HIST: 1985-2014), followed by a mid-century representative period (MC: 2036-2065), and finally, a late(end)-century representative period (EC: 2071-2100). The latter two have been selected under the SSP5-8.5 scenario of CMIP6^{4,25} (see Data Description in Supplementary Section 1.1 and Methods). The methodology used is (...)

6- Line 88: MC and EC have not been discussed before so write them out for the first time, also I would prefer that the timeframes were introduced in the methods section in the introduction.

Please refer to the previous question. MC and EC are now described in the introduction.

7- Line 147-154: could you give some numbers here to make it more quantitative?

The cited paragraph now reads as follows:

The backward Lagrangian approach was used to evaluate future changes in the moisture sources for the IP, our featured target region (see Methods). Figure 1 (left column) show the moisture sources (E-P>0) of the Iberian Peninsula during the historical period for CESM2. In annual terms, it is observed that the precipitation recycling processes (PRPs) —processes that trigger the portion of precipitation whose origin lies in evaporation over the region³²— have a practically homogeneous contribution in the whole peninsula close to 1.4 mm/day. In addition, this figure reveals known seasonal variations²⁰, with a greater contribution in winter from the NATL source reaching as far as the Caribbean Sea (0.4-1.4 mm/day), and predominant influences from the PRPs and the MED in summer and spring (>1.8 mm/day). In autumn, the influence of oceanic sources and PRPs predominates in the east of the IP with values that do not reach 1 mm/day.

8-Figure 1: I would put annual on top in all those figures, and discuss the annual results before moving to the different seasons.

The figures have been updated according to the reviewer's suggestion.

9- After line 186: After reading this results I am very much wondering about the interpretation of these results. Why is the local contribution decreasing in MC and increasing in EC? Do you have any ideas? Putting perspective here would add insights into this paper.

In order to provide a hypothesis for this behaviour, the following paragraph has been included in the latest version of the manuscript:

It is observed a no a priori result concerning the local contribution (particularly associated to the MED, Fig. 4) for regions such as EUWest in AMJ and EUEast in JFM and JAS in which accentuated behaviours are observed for MC with respect to EC or even opposites are observed between both periods. This behaviour would have an explanation of dynamic character, and the explanation can be intuited by considering together the IVT fields (shown in Fig. Sup. 13) together with the changes in Z500 (Fig. Sup. S8). Particularly, in the latter, a latitudinal shift of the geopotential height fields is expected to determine different stability conditions for EC and MC over central Europe. The aforementioned changes in stability conditions would decrease the convergence of moisture flux, and thus precipitation; this could be an explanation for the existence of more pronounced local changes in EC and MC at these seasons.

10- Line 196-197: you use the word reanalysed estimated total precipitation which makes me think of reanalysis products like ERA5 but that's not what you mean here I believe. So what do you mean with reanalysed? Also given the explanation given for PC I would expect a percentage as its mentioned 'a contribution of something to something', but in fact the figure shows absolute numbers which I found confusing. Summarized, I am still not sure what PC exactly entails and I think it is important to make this clear as this paper will be interesting to read for a broad audience who might not all be familiar with the concept of moisture tracking and PCs.

We believe that clarification of what the PCs represent has already been made by answering the reviewer's first question. Regarding the term "reanalysed", it is indeed unnecessary and confusing, so it has been removed from the latest version of the manuscript.

11- Figure 5: In the right column I see this shift in negative and positive contributions over the North Atlantic, especially in winter (JFM, OND), and I was wondering why this happens and if you can discuss (now it is not explicitly mentioned in the text I believe). Is this related to the shift in the location of the storms tracks that you mention earlier?

Certainly, this is the explanation that we consider most reasonable and that, moreover, has already been observed numerous times in the literature (IPCC, 2021; Santos et al., 2016; Harvey et al., 2012). The text has been updated as follows to include a discussion of this:

(...) and the generalized decrease in contribution for the African coast⁴² or the southern Iberian Peninsula^{2,43}. It is also noted that the shift between positive and negative contributions in the North Atlantic — particularly in the winter months— is most likely related to the gradual northward shift of the North Atlantic baroclinic corridor.

The Synthesis section had already discussed this issue in earlier versions of the manuscript:

(...) This will be a consequence of the poleward shift (...) of the storm tracks and the upward expansion of the midlatitude baroclinic regions⁵⁰. This result can be corroborated with the projections of the total precipitation field according to the CESM2 model for IP where a general decrease is expected, being more notable at the end of the century (Pereira et al., 2021; Vautard et al., 2014) (Sup. Fig. 7). This displacement will be clearly related to the shift in the trajectories of extratropical cyclones, since these are the main mechanism of this poleward transport of moisture (Villamil-Otero et al., 2018). This potential shift, which has already been observed in recent decades, is also expected to intensify under less optimistic climate change scenarios (Bender et al., 2012) (...)

12- Line 300-302: Here you discuss the storm tracks, nice, is this the explanation of my question for Figure 5?

Indeed, as explained in the reply to the previous comment; this is the explanation that seems most reasonable.

13- Figure 2, 4 & 6: Is it possible to add an estimate of variability to this plot? As you now have 30 years of tracking per period, the yearly and seasonal variability can be given as a standard deviation for example. An indication of variability would put the MC versus EC into perspective and also gives an indication of the small positive absolute numbers.

As requested by the reviewer, we have performed an analysis of variability and changes in dispersion based on the standard deviation of the daily time series for each of the 30-year ($n = 30 \cdot 365 \approx 11000$ elements) sets of simulations.

The figures obtained from this analysis have been included in the supplementary material, and the following text has been added to the manuscript for discussion of these results:

Having very long time series has also allowed us to analyze changes in the variability of the results presented. In particular, Figure S10 shows the changes relative to the historical period in the standard deviation of the time series of the contribution of the different sources to precipitation in the Iberian Peninsula. In general terms, the variability is also expected to increase, particularly for EC, in values close to 15%. There are some exceptions, such as the case of PRP variability, which is expected to decrease slightly in the winter months for both EC and MC.

Figure S10. Future changes in the yearly and seasonal variability for the relative contribution (in %) of the moisture sources to the Iberian Peninsula. Relative changes are shown for MED, NATL and the PRPs over the four seasons as well as in annual terms both for MC and EC. (INCLUDED IN THE MANUSCRIPT)

Figure S11 shows a variability analysis analogous to that presented in Figure S10 but for the contribution of precipitation from the MED source to its sinks (IP, EUwest, EUeast and NAfrica). In general, an increase in variability with region-dependent values is again observed. For example, the highest values—close to 25% in annual terms for EC—of variability increase are expected for NAfrica, while for IP they remain close to 15%. The case of EUwest is particularly noteworthy, as it shows an increase of 40 percent for JFM, while a decrease in the summer months is expected for the end of the century.

Figure S11. Future changes in the yearly and seasonal variability for the relative contribution of precipitation (in%) to moisture sinks from the Mediterranean source. Percentage of projected future changes in the yearly and seasonal variability for precipitation contribution over EUwest, UEast, NAfrica, and IP associated with the MED source. (INCLUDED IN THE MANUSCRIPT)

Finally, Figure S12 shows analogous results presented for MED in Figure S11, but for the increased variability in the relative contribution to NATL precipitation over its sources (BI, EUwest, IP, West Africa and NAMEast). Again, with some exceptions at the seasonal level, an increase in variability is expected for all sources except West Africa, particularly for EC. NAMEast stands out at the top, with an increase of 30% in annual terms. In the case of West Africa, a decrease in variability is observed, quantified at -10% in annual terms.

Figure S12. Future changes in the yearly and seasonal variability for the the relative contribution of precipitation (in%) to moisture sinks from the North Atlantic source. Percentage future changes in the yearly and seasonal variability for precipitation contribution over: BI, EUwest, IP, WAfrica, and NAMEast associated with the NATL source. (INCLUDED IN THE MANUSCRIPT)

13.a For example in Figure 6c a small number in EUwest annual, is this a small positive number in all 30 years, or is it sometimes also a negative relative contribution?

Since a negative contribution is not possible, we assume that the reviewer is referring to whether the changes in the contribution are small every year, or whether this is the result of a time series that shows great variability, and in which there are years in which the contribution increases and others in which the contribution decreases. It is certainly the latter. As an example, see Figure Rev.5, which shows the time series of the values of P-E>0. There are years in which, for both EC and MC, these values are well above those observed for the time series; while other years are below. In the case of EUwest, taking the whole period as a whole into account, the signal is smoothed down to positive but small net change values.

Figure Rev5.

14- Line 307: To me the Summary is now more of a discussion + conclusion, thus if you keep the discussion into your summary I would choose a different title of this section. An interesting question that popped up in my head when reading the methods is if your travel time of moisture will change into a warmer climate, where the air holds more moisture. And if so, if you need to change your travel time accordingly yes or no. This is not a comment but more of a follow-up question/thought.

As suggested by the reviewer, the title of this section has been changed to "Synthesis", which is certainly much more appropriate.

As for the question regarding the time... this question the reviewer raises is not a trivial one. In fact, it is a question that is under serious discussion these days. Certainly, some studies published in the last decade (some of them by our research group) make us think that with the increase of the amount of water vapor stored in the atmosphere as its temperature increases (given by Clausius-Clapeyron amplification) one could also expect an increase of the residence time. However, this issue is not yet fully clear. In Gimeno et al., (2021) we have made a complete literature review of the results obtained in this respect both by simulations and observations. Although on average it seems that we would expect an increase in residence time of 0.4 days for each degree increase in atmospheric temperature, the reality is that both simulations and observations have obtained very different values; even negative in some cases. In short, we are not yet in a position to alter our simulations by establishing a different residence time depending on the period analyzed. It would not be prudent for the moment to do so, until a more exhaustive investigation is carried out.

15- Line 424: The term WRF-ARW is new for me here, and I don't see how it relates to your work (the -ARW part, what does the abbreviation mean and what is the context?)

The WRF dynamic model has two calculation kernels; WRF-ARW (Advanced Research WRF) and WRF-NMM (Nonhydrostatic Mesoscale Model). Although the first one is the most widely used and developed, it is appropriate to always mention the specific kernel being used, particularly so that no doubts arise among members of the community who also use this model.

OTHER DETAILS

Line 36-37: In addition to warming, an increase in insert(global) mean annual evaporation is expected, remove(which will affect) insert(affecting) the evaporation rates of most oceans.

Line 48-49: With the word this (This projected decrease) it seems like you refer to the previous sentence where you are talking about an increase, which is a bit confusing to read.

Line 70-71: The main means to perform this analysis >The analysis are performed using a Lagrangian

Line 89: region >regions

Line 138; Therefore is an illogic connecting word here for me

Line 153: over the remove(proper) region

Line 195: were remove(tracked) insert(used) to evaluate

Line 206: show results that depend on the season remove(of the year) considered

Line 208: and insert(this is) particularly strong

Line 214: MCs >ECs

Line 266: ... for Figure 3, but now for the NATL

Line 298-299: main responsible >largest contributor to this increase

We thank the reviewer for the time spent in identifying and correcting all our typos. All have been appropriately corrected in the latest version of the manuscript.

References

Brioude, J., et al. The Lagrangian particle dispersion model FLEXPART-WRF version 3.1. *Geosci. Model Dev.* 6, 1889–1904 (2013).

Bender, F. A. M., V. Ramanathan, Tselioudis, G. Changes in extratropical storm track cloudiness 1983–2008: Observational support for a poleward shift. *Climate Dyn.* 38, 2037–2053 (2012).

Gimeno, L., Eiras-Barca, J., Durán-Quesada, A. M., Dominguez, F., van der Ent, R., Sodemann, H., Kirchner, J. W. (2021). The residence time of water vapour in the atmosphere. *Nature Reviews Earth and Environment*, 2(8), 558-569.

Hanna, S. R. Applications in air pollution modeling, in: *Atmospheric Turbulence and Air Pollution Modelling*. Reidel Publishing Company, Dordrecht, Holland, 275–310 (1982).

Harvey, B. J., Shaffrey, L. C., Woollings, T. J., Zappa, G., & Hodges, K. I. How large are projected 21st century storm track changes?. *Geophys. Res. Lett.* 39, L18707(2012).

Hong, S. Y., & Lim, J. O. J. The WRF single-moment 6-class microphysics scheme (WSM6). *Journal of the Korean Meteorological Society* 42, 129–151 (2006).

Hong, S. Y., Noh, Y., & Dudhia, J. A new vertical diffusion package with an explicit treatment of entrainment processes. *Mon. Weather Rev.* 134, 2318–2341 (2006).

IPCC. *Climate Change. The Physical Science Basis. Contribution of Working Group I to the Sixth Assessment Report of the Intergovernmental Panel on Climate Change.* (Cambridge University Press. In Press, 2021).

Iacono, M. J., Delamere, J. S., Mlawer, E. J., Shephard, M. W., Clough, S. A., & Collins, W. D. Radiative forcing by long-lived greenhouse gases: Calculations with the AER radiative transfer models. *J. Geophys. Res.* 113, D13103 (2008).

Jimenez, P. A., Dudhia, J., Gonzalez-Rouco, J. F., Navarro, J., Montavez, J. P., & Garcia-Bustamante, E. A revised scheme for the WRF surface layer formulation. *Mon. Weather Rev.* 140, 898–918 (2012).

Kain, J. S. The Kain–Fritsch Convective Parameterization: An Update. *J. Appl. Meteorol.* 43, 170-181 (2004).

Nieto, R., Ciric, D., Vázquez, M., Liberato, M. L., Gimeno, L. Contribution of the main moisture sources to precipitation during extreme peak precipitation months. *Adv. Water Resour.* 131, 103385 (2019).

Nieto, R., Ciric, D., Vázquez, M., Liberato, M. L., Gimeno, L. Contribution of the main moisture sources to precipitation during extreme peak precipitation months. *Adv. Water Resour.* 131, 103385 (2019).

Ozturk, T., Matte, D., & Christensen, J. H. (2021). Robustness of future atmospheric circulation changes over the EURO-CORDEX domain. *Climate Dynamics*, 1-16.

Pereira, S. C., Carvalho, D., Rocha, A. Temperature and Precipitation Extremes over the Iberian Peninsula under Climate Change Scenarios: A Review. *Climate* 9, 139 (2021).

Shukla, P. R., Skea, J., Calvo Buendia, E., Masson-Delmotte, V., Pörtner, H. O., Roberts, D. C., ... & Malley, J. (2019). IPCC, 2019: Climate Change and Land: an IPCC special report on climate change, desertification, land degradation, sustainable land management, food security, and greenhouse gas fluxes in terrestrial ecosystems.

Tewari, M., Chen, F., Wang, W., Dudhia, J., LeMone, M., Mitchell, K., et al. Implementation and verification of the unified Noah land surface model in the WRF model, 20th Conference on Weather Analysis and Forecasting/16th Conference on Numerical Weather Prediction, Seattle, WA (2004).

Vautard, R., et al. The European climate under a 2 C global warming. *Environ. Res. Lett.* 9, 034006 (2014).

Villamil-Otero, G. A., Zhang, J., He, J., Zhang, X. Role of extratropical cyclones in the recently observed increase in poleward moisture transport into the Arctic Ocean. *Adv. Atmos. Sci.* , 35(1), 85-94 (2018).

REVIEWERS' COMMENTS

Reviewer #2 (Remarks to the Author):

Review round 3

I thank the authors for addressing my comments in this second review round.

The authors have addressed my comment to include the variability of the sources for the two scenarios and have included these figures in the review reply and supplement. I am only confused how it is possible to have negative variabilities? These are for example shown in Figure S10 for IP and JFM, and SSP5-8.5 in AMJ. More negative variabilities appear in Figure S11 and S12. I think it should be clarified how the variability is determined, because if it is expressed with a standard deviation around a mean, I would not expect negative values.

Minor comments

Line 72: write out ARW

Line 79 : to gain greater compression ☐ use better wording? For example: to gain better insights

Line 128: analysed ☐ analysed?

Line 132-133: In contrast, a certain decrease is also 133 observed in the (sub)tropical latitudes of NATL ☐ a certain decrease sounds a bit hand-wavy, is it possible to quantify a bit more?

Line 281: I think it is more useful by how much the percentage has increased then only mentioning the percentage of contribution at EC (because the text does not say the percentage in the historical period)

Line 316: which in annual terms tend to annual each other ☐ tend to cancel each other?

Line 322: with values about from 1-2 mm/day

Line 378-382: This is a very long sentence that is hard to follow and leaves some room for interpretation. Some suggestions:

- Located above ☐ do you mean north of 40degrees?

- Line 381; in this region ☐ do you mean north of 40 degrees or do you mean the North Atlantic with the term 'this region'?

- Line 383: Include a sentence if increased blocking is seen in other GCMs as well or if this is something particular of CESM

More general comments on Synthesis

- As the authors have included some more interpretation into the synthesis to explain their results they also mention one or two times the increase of the 500 hPa geopotential height, and relate this to an increase in anti-cyclonic circulation and thus blocking will be 'greater' (for example line 396). I would personally be a bit hesitant to say so, as you would rather need a local increase of geopotential height (or more wavier jet) to get increases in blocking, rather than an overall increase of geopotential height over the whole area.

- I also notice that in the synthesis there are a lot of references to figures which are located in the supplementary material and thus not directly visible for the reader. Therefore I think it is important to share the main message of this figure in the text to guide the reader a little bit more.

Projected changes in atmospheric moisture transport contributions associated with climate warming in the North Atlantic region
Proposal for publication in Nature: Communications.
ROUND 3 (Aug/Sep 2023)

REVIEWER 2

First, I thank the authors for addressing my comments in this second review round. The authors have addressed my comment to include the variability of the sources for the two scenarios and have included these figures in the review reply and supplement.

We thank the reviewer for the time and effort invested in this third revision of the manuscript. His/her contribution has helped greatly in producing a much higher quality version of the article.

A reply to all his/her suggestions and questions is attached below.

Fernández-Álvarez et al.

MORE GENERAL COMMENTS ON SYNTHESIS

I am only confused how it is possible to have negative variabilities? These are for example shown in Figure S10 for IP and JFM, and SSP5-8.5 in AMJ. More negative variabilities appear in Figure S11 and S12. I think it should be clarified how the variability is determined, because if it is expressed with a standard deviation around a mean, I would not expect negative values

We assume that the reviewer is referring to figures S5, S6 and S7. We understand the reviewer's confusion, and hope to clarify the issue accordingly.

Figures S5, S6 and S7 are not showing standard deviations, but percentage changes in standard deviations. Thus, if the variability decreases over time, the bars will show a negative result; since the standard deviations for the future will be smaller than the standard deviations of the present; so the percentage change will be negative.

In order to avoid possible confusion, the text has been modified as follows:

Having very long time series has also allowed us to analyze changes in the variability of the results presented. In particular, Figure Sup. 5 shows the changes relative to the historical period in the standard deviation of the time series of the contribution of the different sources to precipitation in the Iberian Peninsula. *The cited figure shows the percentage changes in the standard deviation of the future periods with respect to the standard deviation of the historical period. Thus, positive values will indicate an increase in variability; while negative values will show a decrease in variability.* In general terms, the variability is also expected to increase, particularly for EC, in values close to 15%. There are some exceptions, such as the case of PRP variability, which is expected to decrease slightly in the winter months for both EC and MC.

As the authors have included some more interpretation into the synthesis to explain their results they also mention one or two times the increase of the 500 hPa geopotential height, and relate this to an increase in anti-cyclonic circulation and thus blocking will be 'greater' (for example line 396). I would personally be a bit hesitant to say so, as you would rather need a local increase of geopotential height (or more wavier jet) to get increases in blocking, rather than an overall increase of geopotential height over the whole area.

The reviewer is right. The conclusion reached in this respect in the previous version of the manuscript was precipitate. The text has been rewritten as follows:

(...) changes are more noticeable in the eastern region where a weakening of moisture transport towards Eastern Europe is observed. *On the other hand, Z500 fields are expected to strengthen in general, with notable values on a regional scale (e.g. Western European and Mediterranean Sea regions). This behaviour could favour a stronger anticyclonic atmospheric circulation in the future, as well as greater stability. For regions with a local strengthening, it could also lead to an eventual increase in the frequency of blocking flows from their moisture sources.* This evidence indicates that these changes in atmospheric dynamics (...)

I also notice that in the synthesis there are a lot of references to figures which are located in the supplementary material and thus not directly visible for the reader. Therefore I think it is important to share the main message of this figure in the text to guide the reader a little bit more.

The latest version of the manuscript includes different references now explained to the figures in the supplementary material:

This would imply an increased evaporation in the source regions such as the MED and NATL, with an accentuated moisture supply that, at least in part, would end up affecting the IP. Specifically, Fig. Sup. 9 shows a comparison for NATL and the Mediterranean Sea of the evaporation field obtained from WRF-ARW outputs between the historical period and MC and EC for the SSP5-8.5 scenario. This figure shows a remarkable increase in evaporation over the entire Mediterranean Sea, particularly for EC. On the other hand, the NATL region shows a clear differentiated pattern between the Northern and Southern sectors, with evaporation decreasing in the former and increasing in the latter; also particularly noticeable for EC. Moreover, they show a slight latitudinal shift from this source region to the EC (...)

-

In addition, much of this behavior could be attributed to changes in atmospheric dynamics for both MC and EC. Specifically, Fig. Sup. 8 shows the changes in the IVT fields for MC and EC with respect to the historical period for MED under the SSP5-8.5 scenario. This figure shows that for both periods changes in the moisture flux patterns are to be expected.

-

(Fig. Sup. 3, analogous to Fig. Sup. 9 but considering the Z500 field and the outputs of the CESM2 model).

-

(Sup. Fig. 2, analogous to Fig. Sup. 3 but for the precipitation field).

MINOR COMMENTS

- Line 72: write out ARW

The analysis is performed using the Lagrangian dispersion model FLEXPART-WRF initialized with WRF-ARW (WRF with the dynamic core Advanced Research WRF —ARW—) dynamically downscaled outputs. Three representative climatic periods have been used for the comparison.

- Line 79: to gain greater compression ->use better wording? For example: to gain better insights.

Done. Thank you.

- Line 128: analysed ->analysed?

The typo has been corrected.

- Line 132-133: In contrast, a certain decrease is also observed in the (sub)tropical latitudes of NATL ->a certain decrease sounds a bit hand-wavy, is it possible to quantify a bit more?

The text now reads as follows:

In contrast, a decrease of roughly $40 \text{ kg m}^{-1} \text{ s}^{-1}$ is observed in the (sub)tropical latitudes of NATL.

Line 281: I think it is more useful by how much the percentage has increased then only mentioning the percentage of contribution at EC (because the text does not say the percentage in the historical period)

We believe that there has been confusion in the use of the language. What the reviewer is asking for is precisely the information given in the text; the influence of MED on the Iberian Peninsula grows by 20%. To avoid future confusion, the sentence has been rewritten as follows:

In relation to IP, it is observed how MED will continue to be a main source of moisture for this region in the summer months increasing its influence in values close to +20% for EC, relative to the current values³⁵.

Line 316: which in annual terms tend to annual each other ->tend to cancel each other?

Thanks. Fixed now.

Line 322: with values about from 1-2 mm/day

with values ranging from 1 to 2 mm/day.

Line 378-382: This is a very long sentence that is hard to follow and leaves some room for interpretation. Some suggestions:

- Located above do you mean north of 40degrees?

- Line 381; in this region do you mean north of 40 degrees or do you mean the North Atlantic with the term "this region"?

Following the reviewer's suggestion, the text has been rewritten as follows:

Finally, the results show a clear decrease in the contribution of the NATL source in the areas located to the northeast of the North Atlantic near the European shores. These changes may be related to a considerable decrease in evaporation for MC and EC in the aforementioned areas (Fig. Sup. 9). In addition, the increase of the Z500 field could influence the general anticyclonic circulation, as well as an increase of atmospheric stability situations.

- Line 383: Include a sentence if increased blocking is seen in other GCMs as well or if this is something particular of CESM

The following sentence has been included in the latest version of the manuscript:

It is highlighted that with this increase of Z500, a greater increase of blocking situations in future periods can be expected mainly in regions such as Europe and the Mediterranean Sea, where the increase is more noticeable. This is referenced from the results found by Davini and d'Andrea⁴⁷ with different CMIP6 climate models. However, there are numerous discrepancies in the results provided by the different models in the work of these authors; as well as considerable biases in the representation of the present climate⁴⁷ regarding blocking situations. Thus, it is hasty to take these conclusions into account when determining the climate change signal. Instead, a future evaluation of the behavior of this model in the Z500 representation becomes necessary.

References

Davini, P., & d'Andrea, F. (2020). From CMIP3 to CMIP6: Northern Hemisphere atmospheric blocking simulation in present and future climate. *Journal of Climate*, 33(23), 10021-10038.